# A highly accurate risk factor-based XGBoost multiethnic model for identifying patients with skin cancer

Matteo D'Antonio [1] ✉, Wilfredo G. Gonzalez Rivera[1,2], Robert A. Greenes[1,3], Melissa Gymrek[4,5] & Kelly A. Frazer [6,7] ✉

While individuals of European descent have a higher risk of skin cancer, other ancestries tend to have more advanced disease at diagnosis, resulting in outcome disparities. We examine the diverse All of Us dataset and identify genetic ancestries, lifestyle, social determinants of health and PDE5a inhibitor use as independent risk factors for skin cancer. We integrate these risk factors into a highly accurate XGBoost multiethnic model for identifying patients with skin cancer. Analyses of Shapley scores and interactions indicate the presence of strong non-linear associations between age and other risk factors, in particular cancer history, genetic ancestry and annual income, and suggest a stronger dependence on genetic and social determinants in younger individuals. Our XGBoost multiethnic model offers a precision medicine approach for early skin cancer detection in ethnically diverse patients, which could reduce outcome disparities in individuals of non-European ancestries.

Skin cancer is one of the most common cancer types in the United States with an increasing incidence rate[1,2]. While ultraviolet radiation exposure is a well-established risk factor[3], skin cancer is also associated with age, sex, environmental, lifestyle, and socioeconomic factors[4–8], the use of phosphodiesterase type 5 A (PDE5a) inhibitors[9–13], and previous cancer history[14].

Although skin cancer is less common among individuals of non-European ancestries with darker skin tones, it is diagnosed at more advanced stages and is associated with poorer outcomes in these populations[4,5,15]. Differences in early detection rates between individuals of European and non-European ancestries are in part due to skin cancer presentation differences, as melanoma in individuals with darker skin tones often presents in non-sun-exposed areas, such as the palms, soles, mucous membranes, and subungual regions[4], and therefore are less visible. Furthermore, these tumors are easily confused with other lesions, including diabetic ulcers, warts or fungal infections, further complicating the diagnosis[16,17]. Additionally,

dermatologists are better able to diagnose skin cancer on patients with lighter skin compared to patients with darker skin[18]. Studies have shown that individuals of European and non-European ancestries have similar prognoses when skin cancer is detected at an advanced stage[19,20], and hence early diagnosis could potentially alleviate current outcome disparities.

Ultraviolet (UV) light exposure is strongly influenced by multiple factors associated with each individual's lifestyle, such as geographical location, occupational and recreational activities, and tanning behaviors. Social determinants of health (SDOH) risk factors include healthcare access and quality, education access and quality, social and community context, economic stability, and neighborhood and built environment[4]. Three erectile dysfunction drugs, sildenafil (Viagra), tadalafil (Cialis) and avanafil (Stendra), which inhibit PDE5a, have gained substantial attention due to their widespread use and potential causal association with skin cancer risk[9–12,14,21]. The biological premise behind the association between skin cancer and these drugs arise from

¹Department of Medicine, Division of Biomedical Informatics, University of California, San Diego, La Jolla, CA, USA. ²Bioinformatics and Systems Biology Graduate Program, University of California, San Diego, La Jolla, CA, USA. ³Biomedical Informatics, Arizona State University, Phoenix, AZ, USA. ⁴Department of Computer Science and Engineering, University of California San Diego, La Jolla, CA, USA. ⁵Department of Medicine, University of California San Diego, La Jolla, CA, USA. ⁶Department of Pediatrics, University of California San Diego, La Jolla, CA, USA. ⁷Institute of Genomic Medicine, University of California San Diego, La Jolla, CA, USA. ✉e-mail: mdanto@uw.edu; kafrazer@ucsd.edu

their mechanism of action. *PDE5a* expression is downregulated as a result of oncogenic mutations in multiple genes, including *BRAF, KRAS, HRAS, MEK* and *ERK*; and it has been shown that low *PDE5a* expression likely promotes malignant melanoma[22–24]. While these risk factors for skin cancer have been well-established in individuals of European ancestries due to the high incidence rate in this population, their associations with risk in other populations is less well established.

Over the past 20 years, numerous prediction models for skin cancer have been developed, demonstrating the effectiveness of integrating various factors such as lifestyle choices (e.g., sun exposure), genetic predispositions (including family history and phenotypic traits like eye, hair, and skin color), and imaging techniques for identifying nevi, moles, freckles, and sun damage[25–28]. These models had good accuracy (>80%) and not only assist in identifying individuals at risk but, in the case of many imaging prediction tools, also facilitate the direct detection of skin cancer. However, despite their promise, these approaches necessitate substantial amounts of skin-specific data, particularly concerning dermatological visits and imaging procedures. Consequently, they are not easily applicable to broader health databases, such as All of Us or the UK Biobank. Furthermore, they were trained and tested only on non-Hispanic White individuals[25–27].

eXtreme Gradient Boosting (XGBoost) was designed to improve both the performance and speed of machine learning models and offers multiple advantages compared to other machine learning methods that uses tabular data, as it includes built-in Lasso and Ridge regularization, which prevent overfitting and improves model generalization, handles missing data and provides interpretable metrics such as feature importance scores allowing users to understand which features are most influential in the model's predictions. It is currently being used to predict COVID-19, stroke and other health conditions[29,30]. Overall, XGBoost's combination of speed, accuracy, flexibility, and robustness makes it a powerful choice to predict health outcomes.

In this study, to investigate current skin cancer outcome disparities because of diagnosis difficulty in individuals of non-European ancestries, we sought to develop an XGBoost model that can use risk factors to identify individuals with skin cancer regardless of their ancestry. We identify independent associations between skin cancer and genetic and non-genetic risk factors (sex, UV light exposure, lifestyle, socioeconomic factors and PDE5a inhibitors) in a diverse cohort of more than 400,000 Americans in the All of Us Research Program (AoU) database, of whom ~4.4% have been diagnosed with skin cancer. We show that despite a poorer prognosis, individuals of African and Admixed American descent tend to be diagnosed with skin cancer at a significantly earlier age than individuals of European ancestries. We also show that, in genetically admixed individuals, there is a strong linear association between the genotype principal components (PCs) associated with European ancestries and skin cancer risk. We trained XGBoost models only using individuals of non-European ancestries and showed that they have low accuracy in predicting skin cancer patients (F1 mean statistic = 0.067) likely due to low power due to small sample size. Finally, we developed an XGBoost multiethnic model using everyone in the AoU database with complete information on risk factor variables and show that it is highly accurate at identifying skin cancer patients of both European (F1 statistic = 0.903) and non-European (F1 statistic = 0.810) genetic ancestries. Our analyses showed that age and genetics have strong non-linear effects on predictions, which made them important respectively for building the model and for accurate predictions. To investigate the complex relationships between all variables used in the model, we used SHAP scores and interaction values, and showed that the high accuracy of XGBoost is due to the fact that it takes into account non-linear associations between the variables, in particular age, other cancer history, genetic ancestry and annual income.

## Results

### Genetic and predicted ancestries influences skin cancer risk

Since skin cancer risk is extremely variable across populations, we investigated its associations with genetic ancestry determined by AoU from genotype principal component analysis (PCA; Fig. 1A, B). Whole genome sequencing (WGS) data was available for 232,282 participants (including 141,023 females, 60.7%) and for all these individuals genotype PCA and ancestry inference probability was also available through the AoU Workbench. Half of the participants with WGS were of primarily European descent (EUR, 116,847, 50.3%, Table S1), followed by African (AFR, 50,765, 21.9%), American Admixed/Latino (AMR, 39,047, 16.8%), East Asian (EAS, 5225, 2.2%), South Asian (SAS, 2128, 0.92%), and Middle Eastern (MID, 505, 0.22%). The remaining 17,765 could not be annotated to any of these ancestries and were labeled as "Other" (OTH, 7.6%), and include other ancestries or admixed (descendants of individuals from two or more genetic ancestries). Analysis of genotype PC coordinates showed that PC1 separates AFR from the other ancestries, PC2 distinguishes between EAS and EUR, while PC3 and PC4 identify AMR and SAS, respectively (Fig. 1A,B).

Of the AoU participants with WGS, 4.37% (10,151) were diagnosed with skin cancer, the most common being basal cell carcinoma (BCC, 5670), followed by squamous cell carcinoma (SCC, 2,976) and melanoma (2,031). As expected[4], individuals of EUR descent were significantly more likely to develop melanoma than any other ancestry (odds ratio, OR = 6.70, p-value = 7,1e-271, Fisher's exact test, Fig. 1C, Table S2). They were also more likely to be diagnosed with any skin cancer than all the other ancestries (OR = 6.8, p < 1e-300, Fisher's exact test, Fig. 1C, D) and, although they accounted for half of the AoU participants, contributed to 86.4% of all skin cancer diagnoses. The OTH category showed slightly decreased skin cancer risk compared with EUR, but increased risk compared with the other populations, and had the latest age at diagnosis (Figs. 1D, E, S1). While previous studies have shown that Africans are more likely than individuals of European ancestries to be diagnosed with late-stage invasive melanoma[15], our study shows that on average individuals of AFR, AMR and EAS descent tend to be diagnosed with skin cancer at an earlier age than those of EUR descent (AFR 8.3 years earlier, p = 1e-10; AMR 5.08 years earlier, p = 7e-9; EAS 6.0 years earlier, p = 0.004; MID 16.13 years earlier, p = 0.003; SAS 0.8 years earlier, p = 0.7, log-rank test, Fig. 1E, Figure S1, Table S2).

Before further investigating the effects of ancestry on skin cancer, we examined the concordance between predicted continental ancestries and self-reported race and ethnicity. We observed more than 90% concordance for AFR (94.9%), AMR (98.1%), EAS (96.8%), EUR (98.5%) and SAS (95.1%) individuals; more than 85% concordance for MID (85.9%) individuals; but only 12.5% concordance for OTH individuals (Table S1). In general, individuals assigned one of the predicted genetic ancestries had on average >95% of their genome associated with that ancestry regardless of self-reported race/ethnicity (Fig. 2A–D). Of note, the individuals with less than 95% of their genome associated with their genetically assigned ancestries could be stratified based on their self-reported race/ethnicity. AFR individuals that self-report as Hispanic or Latino were on average 9.3% AMR; AMR individuals that self-report as Black (African American) were on average 5.8% AFR; AMR individuals that self-report as White (European) were on average 5.8% EUR and 2.3% EAS (Fig. 2D). For individuals in the OTH category, the ancestral composition of their genomes was even more associated with their self-reported races and ethnicities: self-reported African Americans on average are 47.2% AFR and 18.4% AMR; self-reported Hispanic or Latino are 47.6% AMR, 27.0% AFR and 16.2% EUR; and individuals of self-reported European ancestries are 62.9% EUR, 16.0% AMR and 10.4% MID (Fig. 2D–H).

We divided OTH individuals into three groups according to their self-reported race/ethnicity (Hispanic/Latino, European (Self-reported

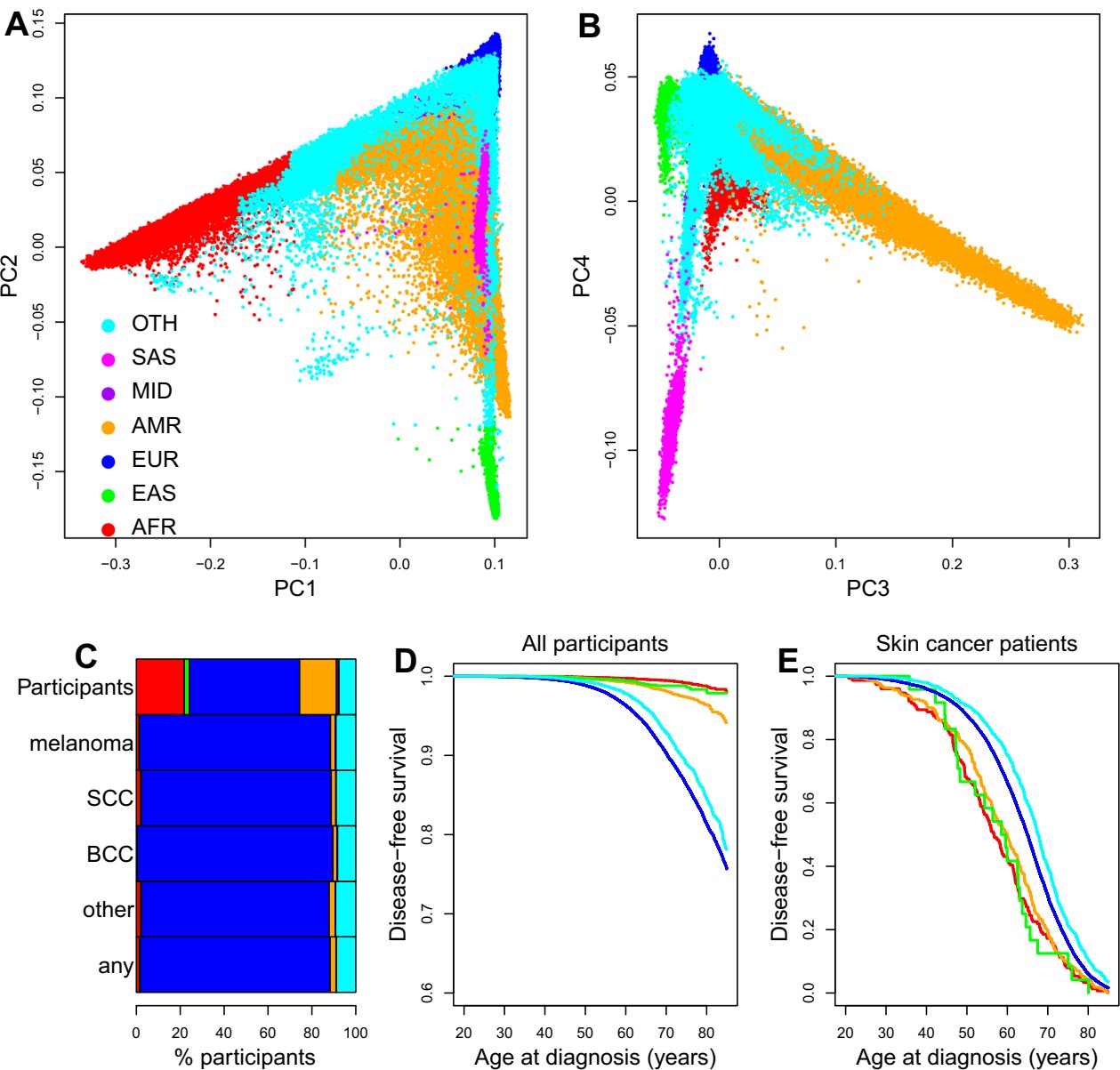

**Fig. 1 | Different skin cancer risk by ancestries. A, B** Scatterplot showing the top genotype PC coordinates for all individuals: A) PC1 and PC2; B) PC3 and PC4. Colors represent genetic ancestries. AFR: individuals of African descent; EAS: East Asian; EUR: European; AMR: Admixed American; MID: Middle Eastern; SAS: South Asian; OTH: all other ancestries and admixed individuals. **C** Proportions of AoU participants from indicated genetic ancestries diagnosed with each cancer category (melanoma, SCC, BCC, other skin cancer type, any type of skin cancer). Top row represents the full AoU cohort. Colors represent genetic ancestries, as shown in (**A**).

**D, E** Survival plots for any type of skin cancer: **D** All participants (healthy and skin cancer patients), EUR and OTH individuals have a greater risk than other genetic ancestries of being diagnosed with any type of skin cancer: X axis indicates age at diagnosis for skin cancer patients; age at last follow-up for all other individuals; **E** only skin cancer patients EUR and OTH individuals tend to be diagnosed at a later age than other populations. Only genetic ancestries with at least 20 skin cancer patients are shown, in accordance with AoU guidelines.

as White) and Admixed (two or more races/ethnicities)) and tested for differences in skin cancer occurrence and age at diagnosis (self-reported groups with less than 20 skin cancer patients were excluded) using a Cox proportional hazard model. We found that OTH individuals who self-reported as White has a higher incidence and later age at diagnosis compared with other OTH individuals who self-reported as Admixed or Hispanic/Latino (p = 5.0e-20 and p = 3.9e-17, respectively, log-rank test, Figs. 3A, B, S2A, B). We also compared OTH individuals who self-report as European with individuals of genetic EUR ancestries, and found that, while they do not display significant differences in incidence (p = 0.82, log-rank test, Figs. 3C, S2C), they are diagnosed at a significantly later age (p = 1.2e-15, log-rank test, Figs. 3D, S2D).

Overall, we show that the associations between predicted genetic ancestries, self-reported race/ethnicity and risk age of diagnosis of skin cancer are highly complex. EUR individuals are more likely to develop skin cancer than individuals in other populations; but AFR, AMR, EAS and MID individuals are more likely to be diagnosed for skin cancer at an earlier age than EUR individuals. While OTH individuals who self-report as European do not display significant differences in incidence compared to individuals with EUR genetic ancestries, they tend to be diagnosed at a significantly later age. Hereafter we represent genetic ancestries using abbreviations (EUR, EAS, MID, AFR, SAS, AMR, OTH), and self-reported races/ethnicities using full names (European, African American, Hispanic or Latino, and Admixed: multiple ethnicities reported).

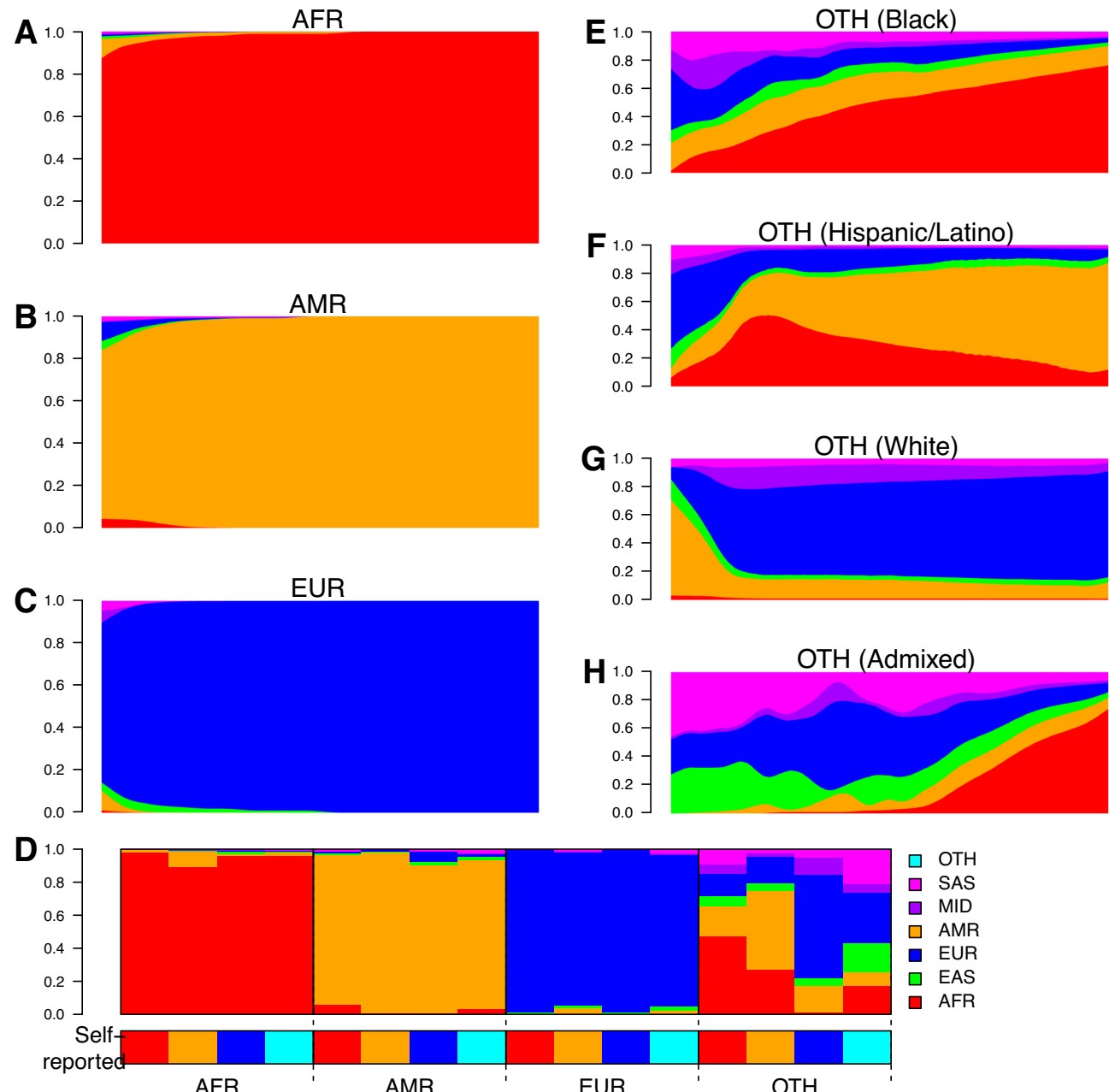

**Fig. 2 | Correspondence between self-reported and genetic ancestry.**
**A**–**C** Barplots showing the ancestry composition of all AoU individuals with the following genetic ancestries: **A** AFR, **B** AMR and (**C**) EUR. **D** Barplots showing the ancestry predictions for AFR, AMR, EUR and OTH individuals based on their self-reported race and ethnicity. Barplots showing the ancestry composition of OTH individuals that self-reported as (**E**) African American (Black), (**F**) Hispanic or Latino (Admixed American), (**G**) White and (**H**) Admixed (more than one race/ethnicity reported).

### Genotype PCs associated with EUR ancestries influences skin cancer risk in OTH and AMR individuals

We examined the association between each AoU genotype PC and skin cancer in non-EUR individuals to determine if risk is influenced by the ancestral composition of their genomes (Fig. 1A, B). Using linear regression, we found significant associations between eight PCs and skin cancer occurrence (Fig. 3E). Seven of the skin cancer-associated PCs separated one of the non-EUR descent from the other non-EUR descent, including two PCs (PC1, PC5) that separated AFR individuals, one PC (PC2) that separated EAS individuals, one PC (PC3) that separated AMR individuals, one PC (PC4) that separated SAS individuals, and three PCs (PC6, PC7, PC8) that separated MID individuals (Fig. S3). One skin cancer risk associated PC (PC14) did not separate the non-EUR populations from each other. Examining

the values of genotype PCs between healthy individuals (defined as all individuals that do not have skin cancer) versus only cancer patients (Fig. 3F, G), we observed that the OTH individuals with cancer have on average a higher value for PC1 and PC2 than when considering healthy OTH individuals (p = 3.9e-221 for PC1 and p = 1.5e-250 for PC2, two-sided t-test, Table S3, Fig. S3). Likewise, AMR individuals with cancer had significantly different PC2, PC3 and PC4 values than healthy AMR individuals (Table S3). These associations largely shift the PC values of OTH and AMR cancer patients towards those of EUR individuals, which suggest that the ancestral composition of the genomes of OTH and AMR individuals, and in particular their proportion of EUR genome, influences their skin cancer risk. Conversely, healthy AFR individuals compared with AFR skin cancer patients were not significantly different for any of the 16

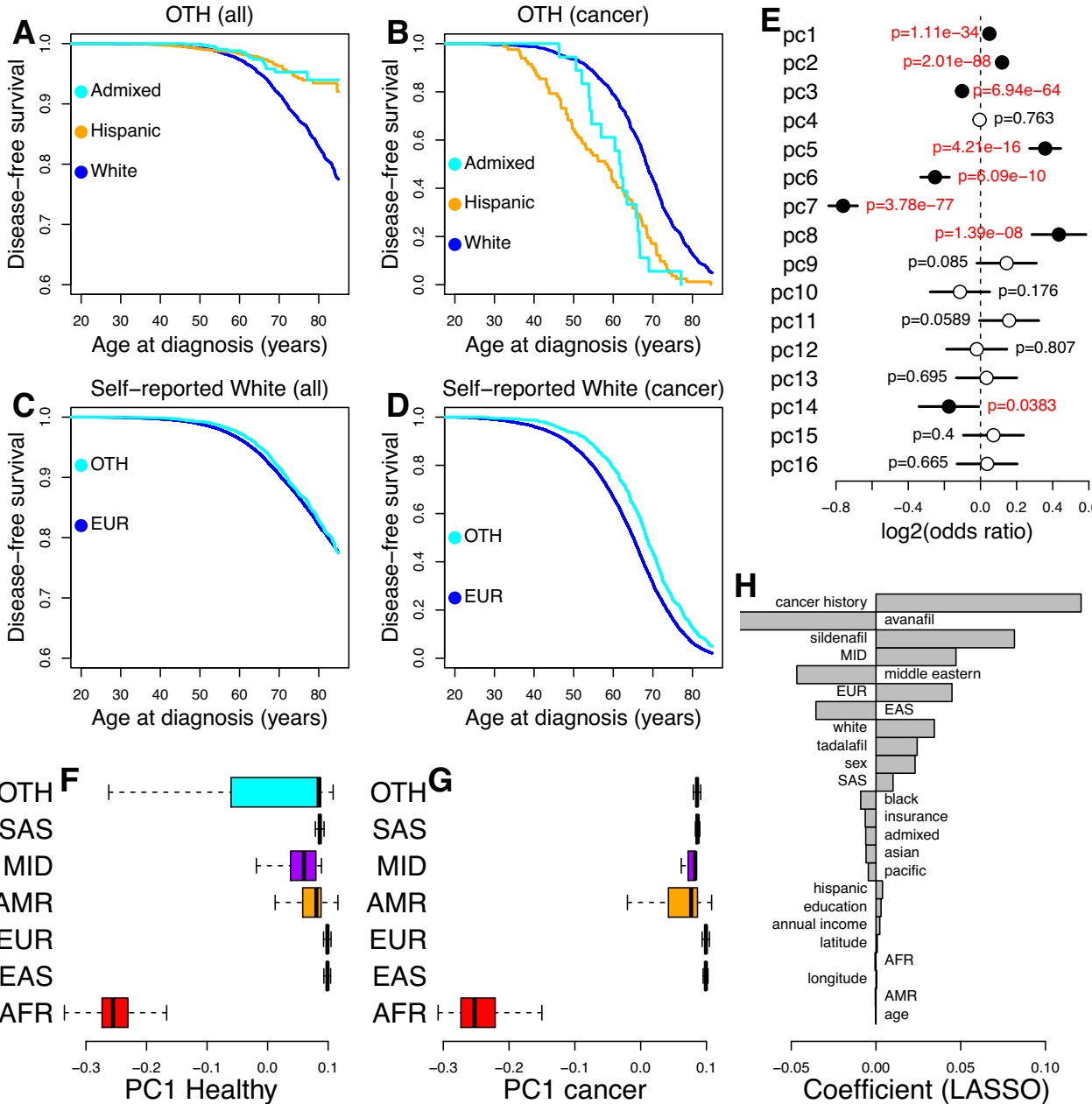

**Fig. 3 | Different skin cancer risk by ancestries. A–D** Survival plots showing for any type of skin cancer: **A** All OTH individuals (healthy and skin cancer patients) grouped according to their self-reported race/ethnicity (Hispanic or Latino, White and Admixed); **B** only OTH skin cancer patients grouped according to their self-reported race/ethnicity (Hispanic or Latino, White and Admixed); **C** All individuals (healthy and skin cancer patients) who self-report as White grouped according to their genetic ancestry (OTH and EUR); **D** only skin cancer patients who self-report as White grouped according to their genetic ancestry (OTH and EUR). Only populations with at least 20 skin cancer patients are shown. In (**A**, **C**), X axis indicates age at diagnosis for skin cancer patients; age at last follow-up for all other individuals. **E** Log2 odds ratio of the association of each genotype PC for being diagnosed with skin cancer in all individuals of non-European ancestry. Segments represent 95% confidence intervals, measured using the *glm* function in R. Filled points represent significant associations (p-value adjusted using Benjamini-Hochberg's method <0.05). Odds ratios, confidence intervals and p-values were calculated using a single logistic regression with outcome = cancer, and the top 16 genotype PCs as covariates. This model was generated removing all individuals of European ancestry. **F**, **G** Boxplots showing the associations between each ancestry and the value of genotype PC1. Healthy individuals (i.e. AoU participants that were not diagnosed with skin cancer, **F** and skin cancer patients (**G**) are shown separately. The central line within each box represents the median, the box edges indicate the 25th and 75th percentiles (interquartile range, IQR), and the whiskers extend to the most extreme data points within 1.5 × IQR from the quartiles. H) Barplots showing the coefficient of each variable used to develop the LASSO logistic regression on OTH individuals. While Avanafil has a strong negative coefficient, it was only prescribed to 46 patients and thus is underpowered. Likewise, individuals that self-identify as Middle Eastern or have MID genetic ancestry included <1000 individuals and are likely underpowered. Abbreviations (EUR, EAS, MID, AFR, SAS, AMR) represent genetic ancestry proportions, while the full names represent self-reported races and ethnicities.

genotype PCs, and likewise for EAS, SAS and MID (Table S3), which suggest that the ancestral composition of their genomes is not a major genetic factor contributing to skin cancer risk in these populations.

To further investigate the associations between the proportions of EUR genetic ancestry in each OTH individual and skin cancer risk, we built a LASSO logistic regression that included the genetic composition of each individual based on genotype PCs, as well as their self-reported race and ethnicity, lifestyle and SDOH. The variables with the strongest influence on skin cancer risk were cancer history and prescription of sildenafil and avanafil, followed by genetic EUR ancestries, self-reported Middle Eastern and self-reported White ancestries (Fig. 3H). These results show that there is a strong linear association between the proportion of EUR ancestries and skin cancer in OTH individuals even when lifestyle and SDOH are taken into account.

In summary, for OTH and AMR individuals, the fraction of EUR admixture influences their skin cancer risk, whereas for the AFR, EAS, SAS and MID populations, the ancestral composition of their genomes is not linearly correlated with skin cancer risk.

## Multiple SDOH and lifestyle factors influence skin cancer risk

It has been hypothesized that differences in skin cancer risk and severity are influenced by non-genetic factors including age, sex, SDOH, drinking, smoking, and PDE5a inhibitor usage[3,4,10,14,15]. Given that individuals of European ancestries are at least eight times more likely to be diagnosed than other populations, we grouped individuals into "All" populations, EUR only, and non-EUR only, and performed logistic regression analyses to test the association between risk for any skin cancer and multiple factors independently, including: cancer history; age; sex; SDOH (annual income, insurance status, education level, living situation, latitude, and longitude); lifestyle (daily and yearly alcohol intake and smoking); self-reported races and ethnicities (Hispanic, African, European, Admixed, Asian, Middle Eastern, Pacific Islander); genetics (top 16 genotype PCs); and prescription of PDE5a inhibitors (avanafil, sildenafil and tadalafil). Upon testing each single factor independently, we observed that almost all variables were significantly associated with increased or decreased risk for any skin cancer and had similar enrichments between EUR and non-EUR individuals (Fig. 4A–C, Table S4). When examining all participants all factors but avanafil, self-reported Pacific Islander, and PC11 were significantly associated. The main difference between EUR and non-EUR groups was represented by genotype PCs: given that the non-EUR group was more heterogeneous than the EUR group, a larger proportion of genotype PCs was significant.

To determine which of the tested variables are independently associated with skin cancer, we performed a logistic regression analysis combining all variables into a single model. Most variables, other than age, self-reported race/ethnicity, avanafil use and genotype PCs 3-16, were independently associated with risk for any skin cancer in the multiethnic model that included all participants (Fig. 4D, Table S4). However, when we divided into EUR and non-EUR individuals, we observed multiple differences between the two groups: while cancer history, sex, daily alcohol intake, smoking daily, and longitude were significant for both, the other SDOH and lifestyle factors, the two PDE5a inhibitors (sildenafil: $p = 45e-6$; and tadalafil: $p = 2.7e-4$) and genotype PC2 were significant only for individuals of European ancestries, whereas genotype PC1 and genotype PC3 were significant only for individuals of non-European ancestries (Fig. 4E, F, Table S4). Since PDE5a inhibitors are mainly prescribed for erectile dysfunction (ED), we repeated this analysis only in male individuals, and observed similar trends, with both sildenafil and tadalafil being independently associated only in individuals of European ancestries ($p = 1.1e-6$ and $p = 6.3e-5$, respectively, Table S4).

Since it has been shown that wealthier individuals often exhibit a lower incidence of more invasive skin cancers[31,32], we investigated the associations between annual income and skin cancer incidence in EUR individuals. We found that wealthier EUR individuals are more likely to be diagnosed with skin cancer ($p = 1.0e-82$, Cox proportional hazard test), are diagnosed earlier ($p = 2.7e-34$, Cox proportional hazard test) and, among skin cancer patients, have an increased survival rate ($p = 3.5e-12$, linear regression, Fig. S4), with skin cancer patients that earn <\$10,000 per year being 7.6 times more likely to die at a younger age than skin cancer patients that earn >\$200,000 per year. These results support previous observations that wealthier individuals have a more favorable skin cancer prognosis.

Overall, these results show that in the multiethnic logistic regression model multiple factors, including cancer history, sex, SDOH, lifestyle, PDE5a inhibitor use and genetic ancestries, independently affect a person's risk to develop any skin cancer. Of note these observations indicate that the information provided by self-reported race/ethnicity about skin cancer risk are captured by other variables.

## Logistic regression models poorly predict individuals with skin cancer

Given that skin cancer is difficult to detect in non-EUR populations and hence detected at more advanced and invasive stages[33], we assessed the utility of logistic regression models trained with associated risk factors to predict individuals with skin cancer. Since not all participants provided survey feedback or had genetic information, we first selected 179,094 participants (45.7%) who had all values for the variables described in Fig. 4, which we split into 80% training (N = 142,252) and 20% test (N = 35,563) sets. We trained logistic regression models using all variables as input. We trained 45 models using combinations of the five cancer categories (any skin cancer, melanoma, SCC, BCC, and other skin cancers), three genetic population groups (All participants, EUR, and non-EUR) and sexes (all (both sexes), males, and females). All models showed high sensitivity (mean = 0.795), specificity (mean = 0.745), negative predictive value (NPV, mean = 0.992) and recall (mean = 0.795), but low positive predicted value (PPV, mean = 0.057), precision (mean = 0.057) and F1 statistic, which provides a balance between precision and recall, especially with imbalanced datasets (mean = 0.103, Fig. S5, Table S5).

In summary, these data show that the logistic regression models were not able to predict which individuals had skin cancer with high PPV and precision. Based on the complex relationships between genetic, lifestyle and socioeconomic risk factors and their associations with skin cancer, we hypothesized that XGBoost, which is a machine learning model that considers non-linear associations, would be able to better predict whether an individual has skin cancer.

## XGBoost multiethnic model accurately predicts individuals with skin cancer

We set out to use XGBoost to develop a high-accuracy machine learning model capable of utilizing genetic and non-genetic risk factors to predict which individuals have skin cancer with high accuracy regardless of their genetic ancestry. We selected the same 179,094 AoU participants used for the logistic regression models above who had all values for the variables described in Fig. 4 (including genotype information, SDOH and lifestyle), and split them into the same 80% training (N = 142,252) and 20% test (N = 35,563) sets. We optimized 45 XGBoost models using combinations of the five cancer categories (any skin cancer, melanoma, SCC, BCC, and other skin cancers), three genetic population groups (all participants, EUR, and non-EUR) and sexes (all, males, and females). Compared with the logistic regression models, we observed overall increased sensitivity (mean = 0.976, cross-validation; mean = 0.988, validation set), specificity (mean = 0.907, cross-validation; mean = 0.989, validation set) and recall (mean = 0.974, cross-validation; mean = 0.989, validation set, Table S6). While both sensitivity and specificity, as well as NPV (mean = 0.999, cross-

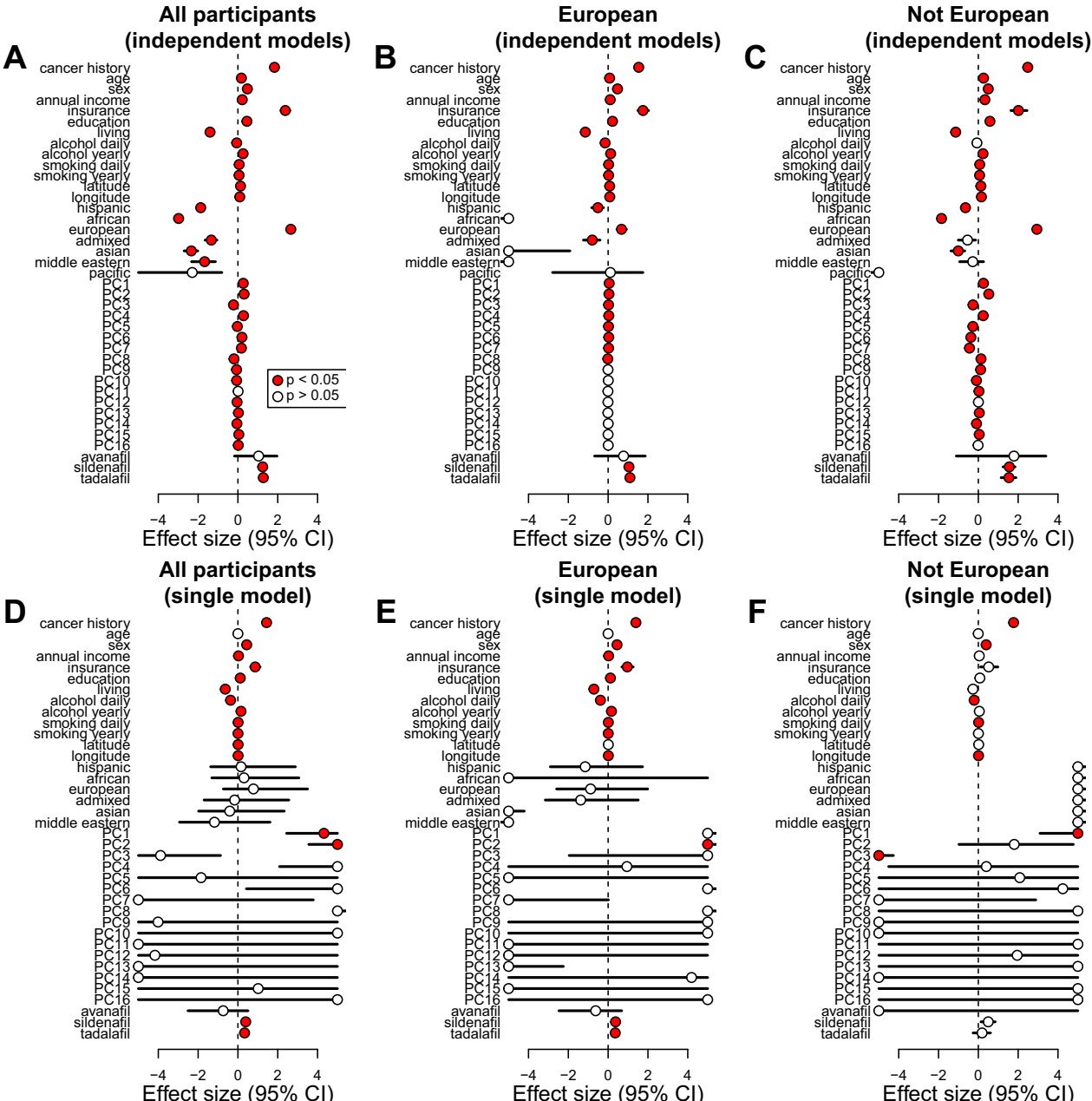

**Fig. 4 | Associations between SDOH, lifestyle, ancestry and skin cancer.** Plots showing the effect size and 95% confidence interval for (**A–C**) logistic regressions on each independent variable for (**A**) all participants, (**B**) individuals of genetic European ancestry and (**C**) individuals of non-European ancestry; and (**D–F**) logistic regressions on all variables combined for (**D**) all participants, (**E**) individuals of European ancestry and (**F**) individuals of non-European ancestry. Significant signals (Benjamini-Hochberg's adjusted p-values < 0.05) are highlighted in red. Covariates with effect size >4 and < −4 were set to 4 and −4 respectively for visualization purposes. P-values associated with the analyses reported in this figure are shown in Table S4.

validation; mean = 0.999, validation set), were consistently high, the precision and F1 statistic were highly variable across the 45 models (Fig. 5A). The highest F1 statistic was for the model trained to predict any skin cancer occurrence, in the EUR group, in both sexes (F1 statistic = 0.936, cross-validation; F1 statistic = 0.903, validation set, Fig. 5A, Table S6); but the F1 statistic of this model was only marginally better than the model trained to predict any skin cancer occurrence, in the All participants group, in both sexes (F1 = 0.892, cross-validation; F1 statistic = 0.892, validation set, Table S6), which we hereafter refer to as "the XGBoost multiethnic model". Conversely, the 15 models trained only using non-EUR individuals had large numbers of false positives and small numbers of true positives, which resulted in low

F1 statistics (mean = 0.067, Table S6). The poor performance of the non-EUR models was likely due to the fact that they were underpowered due to the relatively small number of non-EUR cancer patients. Overall, these results suggest that the XGBoost multiethnic model could potentially improve the prediction of skin cancer incidence in individuals of non-European ancestries.

To evaluate the impact of specific variables on skin cancer predictions, we analyzed changes in prediction accuracy by comparing modified versions of the XGBoost multiethnic model where each class of risk variables was systematically excluded during training. We trained models using the same 80% training and 20% test sets as the 45 models described above by removing each of the nine classes of risk

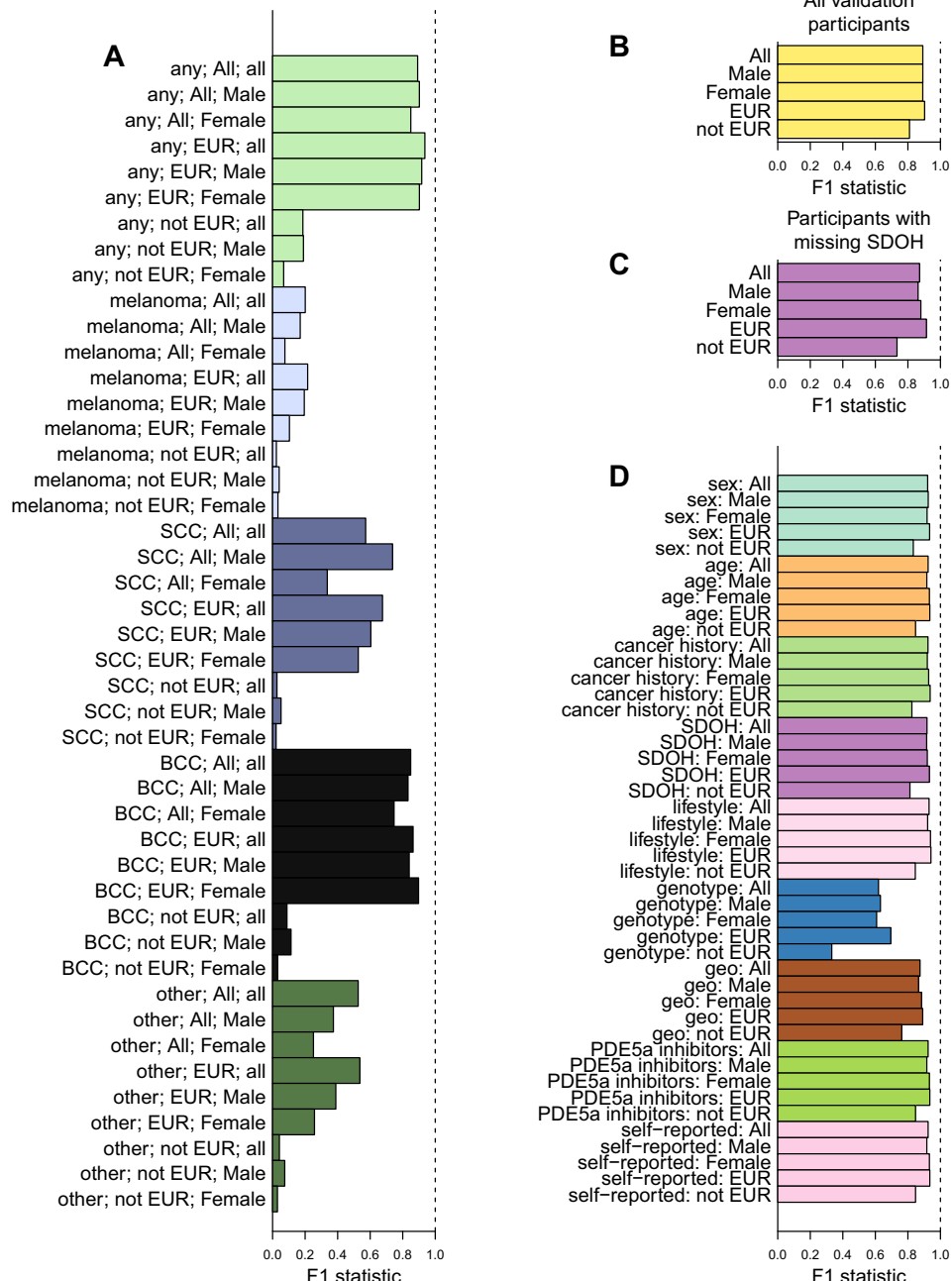

**Fig. 5 | Stratification of individuals with skin cancer using XGBoost. A** Barplots showing the F1 statistic of predictions across all 45 models which consist of combinations of the five skin cancer categories (any, melanoma, SCC, BCC, other), three genetic populations groups (EUR, not EUR, All – both EUR and not EUR), and sexes (Male, Female, all - both Males and Females). F1 statistics were calculated on the cross-validated training set. **B**, **C**) Barplots showing the F1 statistic using the XGBoost multiethnic model (any skin cancer, all participants and both sexes) for AoU participants: **B** F1 statistic for all validation individuals that have all information (see Table 1); and **C** F1 statistic for all AoU participants missing SDOH information. The F1 statistic results are shown when considering All individuals, Males only,

Females only, individuals of European ancestry only, and individuals of non-European ancestry only. **D** Barplots showing the F1 statistic of validation individuals using the XGBoost multiethnic model (any skin cancer, all participants and both sexes) when one variable (sex, age, cancer history, SDOH, lifestyle, genotype PCs, geo = longitude and latitude, PDE5a inhibitors) at a time was removed. Each color represents the removal of a single variable. The F1 statistic results are shown when considering All participants, Males only, Females only, individuals of European ancestry only, and individuals of non-European ancestry only. The columns associated with each missing variable were set to NA.

---

variables one at a time (age, sex, cancer history, SDOH, lifestyle, genotype PCs, longitude and latitude, PDE5A inhibitors, and self-reported race and ethnicity) and evaluated the prediction accuracy in all participants, Males, Females, individuals of European ancestries, and individuals of non-European ancestries (Fig. S6, Table S7). Removing eight of the risk variables (sex, cancer history, SDOH, lifestyle, genotype PCs, geo = longitude and latitude, PDE5a inhibitors, and predicted ancestry)

one at a time from the XGBoost multiethnic model had minimal effects on its accuracy (mean F1 statistic = 0.807 across eight variables, Fig. S6, Table S7). However, removing age resulted in a large decrease in predictive performance (F1 statistic = 0.217, Fig. S6, Table S7). Specifically, we observed a 29-fold increase in the number of false positives compared with the full XGBoost multiethnic model (11,369 compared with 390 in the full multiethnic model), who were on average 10.1 years

**Table 1 | Prediction accuracy in validation set of individuals**

| Dataset | Positive predictive value | Negative predictive value | Sensitivity | Specificity | Precision | Recall | F1 |
|---|---|---|---|---|---|---|---|
| All | 0.80882353 | 0.99967187 | 0.99337748 | 0.98849625 | 0.80882353 | 0.99337748 | 0.89165091 |
| Male | 0.80712166 | 0.99976744 | 0.996337 | 0.98510541 | 0.80712166 | 0.996337 | 0.89180328 |
| Female | 0.81049563 | 0.99961208 | 0.99049881 | 0.99062951 | 0.81049563 | 0.99049881 | 0.89150187 |
| EUR | 0.82481752 | 0.99978101 | 0.99728445 | 0.98320233 | 0.82481752 | 0.99728445 | 0.90288875 |
| not EUR | 0.6988417 | 0.99954119 | 0.96276596 | 0.99491127 | 0.6988417 | 0.96276596 | 0.8098434 |

Prediction accuracy of XGBoost multiethnic model (any skin cancer, all participants and both sexes) in the validation set of individuals. Shown are PPV, NPV, sensitivity, specificity, precision, recall and F1 statistic. The results are shown when considering All participants, Males only, Females only, EUR only, and non-EUR only.

older than the true negatives (95% CI 9.76-10.47, p < 1e-300, t-test). These findings indicate that missing age information while building the model results in the over-prediction of skin cancer in older individuals. This is consistent with the observation that in the XGBoost multiethnic model age is the feature with the highest importance (gain = 0.712, Table S8); but given the lack an independent association between age and skin cancer in the logistic regression models (Fig. 4D–F) our findings suggest that there could be a non-linear associations between age and other risk variables.

To further characterize each variable's effect on skin cancer prediction, we next examined the ability of the XGBoost multiethnic model to identify skin cancer patients in the test set of participants (Table 1). Considering all test individuals (N = 35,563) the XGBoost multiethnic model had a strong F1 statistic (0.892) and precision (0.809, Fig. 5B, Table 1). For the 1661 individuals of European ancestries with any skin cancer in the test set, there were less than 20 false negatives using the full multiethnic model (F1 statistic = 0.903, Table 1), and for the 188 individuals of non-European ancestries with any skin cancer there also was less than 20 false negatives and hence a high F1 statistic (0.810, Table 1). This analysis shows that the XGBoost multiethnic model applied to the test set of individuals was highly accurate at identifying both individuals of European and non-European ancestries with skin cancer, indicating its utility as a prediction tool for all populations.

### Non-linear associations between age and other variables influence multiethnic predictions

While the XGBoost multiethnic model had relatively few false negatives in the test set of participants, there were hundreds of false positives. We explored whether false positive individuals exhibited distinct characteristics compared to those correctly classified as skin cancer patients. To do this, we conducted a logistic regression analysis in the EUR group, with the outcome categorized as either true positive (n = 1644) or false positive (n = 353), using all variables from the XGBoost multiethnic model as covariates. The analysis revealed that most risk variables were not significantly different between the true positives and false positives; however, false positives were generally younger (p = 3.7e-15) and were less likely to have a history of cancer (p = 2.1e-27, Table S9). These findings suggest that younger individuals with many risk factors (excluding a history of cancer) tend to be incorrectly predicted to have skin cancer.

Since a quarter of the AoU participants (55,459, 23.2%) were not used in the training or test sets because they had missing SDOH or lifestyle from the AoU surveys, we investigated how the accuracy of the XGBoost multiethnic model was impacted with missing SDOH information. Using this "missing feature" set, we observed that missing SDOH data slightly decreased prediction accuracy in both the All group (F1 statistic = 0.872; Fig. 5C, Table S10) and non-EUR group (F1 statistic = 0.733, Fig. 5C, Table S10), suggesting that the XGBoost multiethnic model works well even when the test participants have missing SDOH data. To further understand the effects of missing data

on prediction accuracy of the XGBoost multiethnic model, from the 35,563 test participants, we removed each set of risk variables one at a time and found that only removing genotype PC information impacted the accuracy of the XGBoost multiethnic model (F1 = 0.619, Fig. 5D, Table S10). We also tested whether this strong reliance of the model on age is due to the fact that this variable is defined in different ways for skin cancer patients (age at diagnosis) and other individuals (current age). We built a model using current age for all individuals and obtained very high F1 statistic (0.909). These findings show that the XGBoost multiethnic model is robust to the presence of missing variable information, suggesting that it may provide accurate predictions in real-world scenarios, where some patients will have missing information. It also shows that the genotype PCs of each individual are important to make correct predictions, as the accuracy of the model is decreased by more than 25% when information about genetics is removed.

Altogether, these results show that implementing a multiethnic non-linear machine learning model that includes genetics, lifestyle and SDOH results in the accurate prediction of skin cancer occurrence in both EUR individuals, which account for the vast majority of skin cancer cases, and other ancestries, which alone would be too underpowered to result in an accurate prediction. Removing the genetic component from the predictions results in a significant decrease in the model performance, suggesting that there are genetic features that influence the model that cannot be captured by any other variable.

### Age and genotype PCs have strong non-linear associations with the prediction model

We further investigated the contributions of each variable to the model to better understand why age has such a strong influence on accuracy when building the model and missing genotype PCs results in strongly decreased model prediction accuracy. We assessed SHAP scores, which quantify the contribution of each feature to each prediction, and interactions between variables and, as expected, we found that age had the strongest contribution (Fig. S7), even when testing the predictions only in the top or bottom quartile values for age (Fig. 6A–C). Interestingly, age had a strong positive SHAP value in both the top and bottom quartiles, but had a larger contribution to the cumulative prediction in the top quartile. In older individuals (top quartile), in addition to age the strongest contributors to the model were self-reported White ancestries (negative), cancer history (positive) and PC2 (negative); whereas in younger individuals (bottom quartile) after age, PC2 had the strongest positive contribution, followed by a combination of the 30 variables with the lowest SHAP scores, and then cancer history. In total, these observations suggest that: 1) age has a strong non-linear effect on predictions (Fig. S7); and 2) the effects of genetics on the model change with age, with PC2 having a strong positive contribution in younger individuals and a negative contribution in older individuals. To further study the effect of genetics on predictions we investigated individuals in the top and

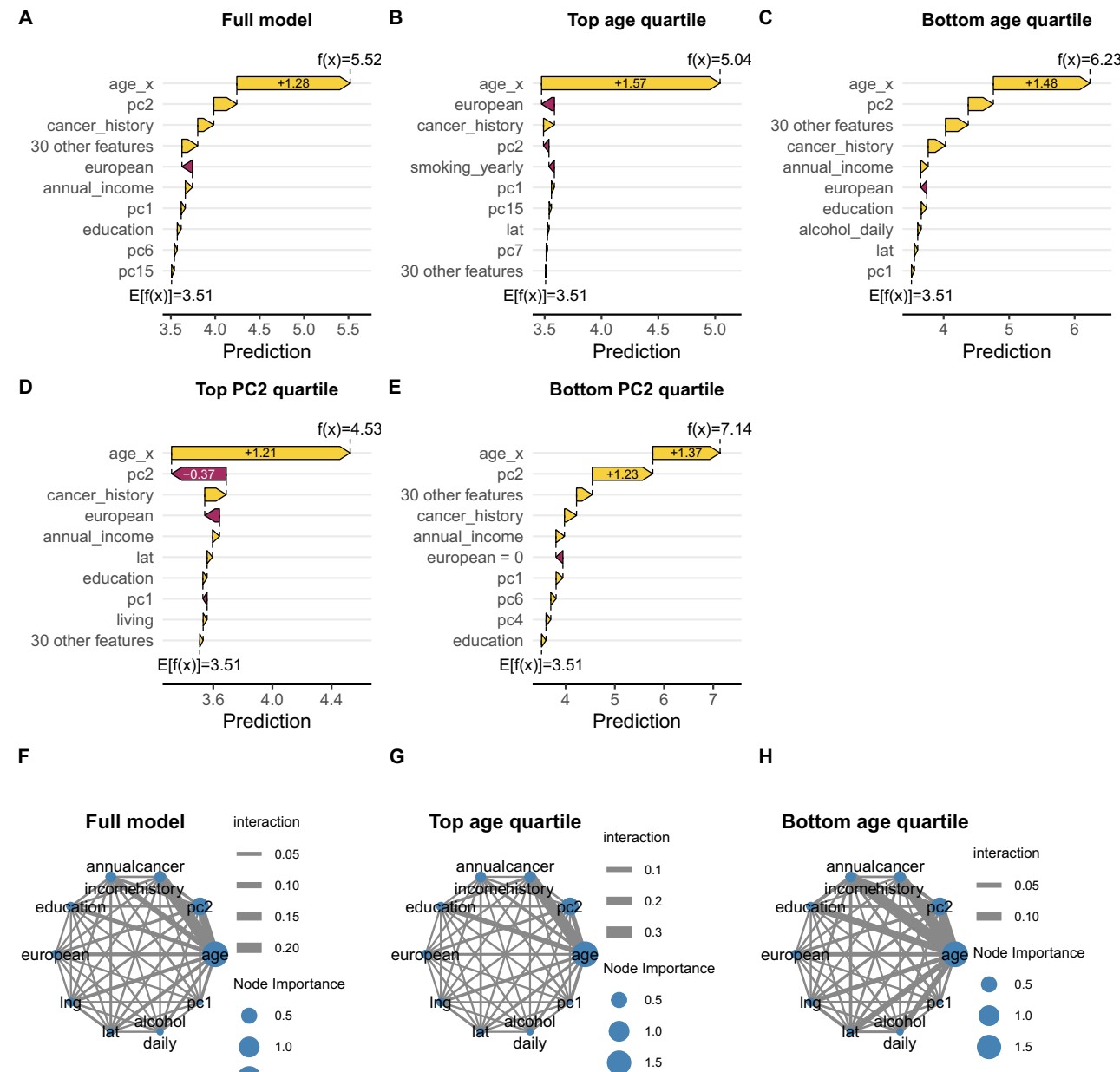

**Fig. 6 | Non-linear associations between variables in the XGBoost model.**
**A–C** SHAP waterfall plots showing how each feature positively or negatively contributes to each prediction: **A** full dataset; **B** only individuals in the top quartile for age; **C** only individuals in the bottom quartile for age; **D** only individuals in the top quartile for PC2; **E** only individuals in the bottom quartile for PC2. The plots were generated using the *sv_waterfall* function in shapviz 0.9.7 in R. The plot shows how each feature contributes to the model's prediction for the sets of representative individuals described above, starting from the baseline prediction (mean model output). Bars indicate SHAP values, with positive contributions (increasing the prediction) shown in yellow and negative contributions (decreasing the prediction) in purple. Features are ordered by the absolute magnitude of their impact on the prediction. The final predicted value (right-most point) reflects the cumulative

effect of all feature contributions. Plots showing the SHAP interactions between the ten variables that have the strongest effect on the model: **F** full dataset; **G** only individuals in the top quartile for age; **H** only individuals in the bottom quartile for age. The interactions were calculated using the *shap.prep.interaction* function in SHAPforxgboost 1.0.3 in R. Node size is proportional to feature importance (Table S8), while vertex width is proportional to the interaction value between each pair of features. We used SHAP interaction values to examine whether the predictive influence of certain features depended on other features. For example, the interaction between age and cancer history showed that the effect of cancer history was more pronounced in older individuals, indicating a non-additive, synergistic relationship between these features. Conversely, in younger individuals, the interaction between age, PC2 and annual income was more pronounced.

bottom quartiles for PC2, and found that for individuals in the top PC2 quartile, age is the strongest contributor to the model and PC2 has a negative association, while for individuals in the bottom PC2 quartile, the SHAP values of age and PC2 are similar. These observations indicate that, for both older individuals and individuals with high fractions of EUR ancestries (high PC2), age is the strongest positive contributor to the model and PC2 values are a negative predictor, while for both

younger individuals and individuals with low fractions of EUR ancestries (low PC2), the PC2 values are important positive predictors of skin cancer (Fig. 6D, E).

We also used SHAP interaction values to examine whether the predictive influence of certain variables depended on other features (Fig. 6F–H). Investigating the interaction between age and cancer history showed that the effect of cancer history was more pronounced in older

individuals, indicating a non-additive, synergistic relationship between these features. Conversely, in younger individuals, the interaction between age, PC2 and annual income was more pronounced, suggesting a stronger association between skin cancer risk, genetics and wealth in younger individuals. This indicates non-additive relationships, where the joint influence of these features differs from the sum of their independent effects, highlighting a conditional dependency that may be clinically relevant. These results also confirm that age and genetics have strong non-linear effects on predictions, which make them important respectively for building the model and for accurate predictions.

## Discussion

Understanding associations between genetic predispositions, lifestyle choices, SDOH and the onset of skin cancer holds profound significance for clinicians and patients alike. With the proliferation of extensive and diverse datasets encompassing Electronic Health Record (EHR) data, genotype information, and socioeconomic profiles, elucidating these relationships and leveraging them to enhance clinical decision-making and patient education is increasingly within reach.

In this study, we leveraged the expansive All of Us (AoU) dataset to delve into the intricate relationships between skin cancer, genetic predispositions, lifestyle factors, SDOH, and the utilization of PDE5a inhibitors. Using these relationships, we developed a machine learning XGBoost model, finely tuned for diagnosing skin cancer across diverse population cohorts. Notably, while our model demonstrates superior predictive capabilities for skin cancer occurrence among individuals of European ancestries compared to non-European ancestries, its multi-ethnic framework significantly enhances predictions across all demographic groups, surpassing models solely built on non-European populations. Furthermore, while logistic regression analysis indicated that age was not independently associated with skin cancer occurrence, its exclusion from the training of our XGBoost multiethnic model led to a notable decline in prediction accuracy. This is in general expected in cancer studies, as age is a major risk factor for most cancers. The discrepancy in the association with age between logistic regression and XGBoost can be explained by several factors, including: 1) model type and complexity: given that XGBoost is a tree-based ensemble method that captures non-linear relationships between features and the target variable, it accounts for complex interaction between variables, whereas logistic regression assumes that each predictor has a direct and additive effect on the log-odds of the outcome; 2) multicollinearity: if age is correlated with other variables, logistic regression may assign the effect to those other correlated variables, reducing the apparent significance of age; in contrast, XGBoost can better handle multicollinearity due to its tree-based structure, which does not rely on independent variable assumptions; and 3) data scaling and feature transformation: if age requires any transformation (e.g., quadratic or log transformation), it might not appear significant; XGBoost, on the other hand, doesn't require feature scaling and can automatically handle non-linearity in the data. The observation that missing age information while building the model results in the over-prediction of skin cancer in older individuals suggests that each additional risk factor contributes incrementally to prediction accuracy, providing information that is correlated with age. For example, there are strong associations between age and lifestyle factors, SDOH and using PDE5a inhibitors. Consequently, the absence of any single risk factor while building the model other than age does not significantly impair the model's predictive performance. This phenomenon sheds light on the fact that missing variable data (other than genotype PCs) for test participants does not substantially compromise prediction accuracy, as the model relies on a combination of statistically independent yet correlated risk factors to make accurate predictions. The fact that current age has a strong influence on building the XGBoost multiethnic model limits its application for predicting future skin cancer risk, but makes it an ideal tool for clinicians to use to identify which patients should undergo whole-body skin cancer examines by trained dermatologists. Conversely, predictions are not strongly affected by age, as removing this variable from the test set does not decrease the model's accuracy. The only variable set whose removal strongly decreased the model's accuracy in the test set was genetics, suggesting that, while the non-linear relationships between all the other variables make them redundant for predictions but not for building the model, there is a genetic component that cannot be capture by anything else, making genetic information necessary for accurate predictions of skin cancer. In general, three main factors influence skin cancer risk: genetics, environment (intended as both lifestyle and SDOH), and the use of PDE5a inhibitors. In this study, we show how these factors influence skin cancer, as well as how they are important for skin cancer predictions:

1. *Genetics*. Skin cancer predominantly affects individuals of European ancestries, but individuals of African and Asian ancestries tend to present with more advanced stages and experience poorer survival rates[4,5]. These disparities have been hypothesized to be largely attributed to variations in skin pigmentation (melanin levels) and vitamin D production, which are genetically determined[34–37]. In this study, we adopted a global genotype PCA approach to incorporate genetic data into our skin cancer prediction model: by using this approach, there is no need to divide populations into distinct categorical variables, but we can analyze all participants, including Admixed individuals that frequently are excluded from genetic studies, in a single model[38]. Interestingly, Admixed individuals who self-report as White have a higher age at diagnosis than individuals with European genetic ancestries, even when accounting for lifestyle and SDOH. Our results confirm the findings from previous studies[39], which demonstrated the importance of adjusting for environmental factors when investigating disease risk in diverse and admixed populations. We show that, for individuals of OTH (which include individuals with multiple genetic ancestries) and AMR ancestries (Admixed Americans, who have substantial contributions from Native American, European, and African ancestry), skin cancer patients were located closer to EUR individuals in the genotype PCA space than healthy individuals. We show that, by combining all individuals into a single model, prediction accuracy greatly improves for individuals of non-European ancestries. However, prediction accuracy is impacted when genotype PCs are not included for individuals, which shows that genetic information plays an important role in the robust skin cancer predictions by the XGBoost multiethnic model. While overall our model shows improvements in predicting individuals of genetic non-EUR ancestries, further studies should be performed to further characterize other potential biases and ensure model fairness.

2. *Lifestyle and SDOH*: Assessing the impact of non-genetic factors on skin cancer risk presents inherent complexity, given the intricate web of interconnected variables at play. For example, extensive research has illuminated the associations between healthcare access, wealth, education, and health insurance coverage. Wealthier individuals often exhibit a lower incidence of late-stage melanoma, attributed to heightened health awareness, and lifestyle, such as avoiding smoking[13,40], and the increased likelihood of possessing health insurance and receiving cancer diagnoses at earlier, more treatable stages[31,32]. Furthermore, wealth is strongly linked to early detection cancer screenings, which facilitate timely diagnoses, interventions, and improved prognoses overall. These findings underscore the profound influence of socioeconomic factors on skin cancer outcomes, highlighting the need for comprehensive strategies to address disparities and enhance early detection efforts[41–44]. Given the interconnectedness of lifestyle and socioeconomic factors, excluding any single variable from our predictive model typically yields negligible effects on overall predictions. Because of this partial redundancy, our observations indicate that the XGBoost

multiethnic model proposed here exhibits robust performance even when confronted with partially incomplete patient records. This underscores the critical importance of integrating comprehensive lifestyle, health, and socioeconomic data to attain consistently high model accuracy.

3. *Use of PDE5a inhibitors*: Multiple studies have shown an association between the use of PDE5a inhibitors, particularly among individuals with a history of cancer, and an elevated risk of skin cancer[9–12,14,21]. This association was discovered after the biological significance of downregulating *PDE5a* expression by cancer driver mutations in genes within the RAS/BRAF/MEK/ERK pathway was realized[22–24]. However, a causal link between PDE5a inhibitors and skin cancer is difficult to establish because of the interconnectedness of lifestyle and socioeconomic factors that we show in our study. Notably, affluent individuals, who are more inclined to use PDE5a inhibitors, often exhibit higher levels of recreational sun exposure and tanning, both recognized risk factors for skin cancer[13]. In this study, while our focus did not encompass establishing causality between PDE5a inhibitors and skin cancer, we show that PDE5a inhibitor use is independently associated with skin cancer risk in individuals of European ancestries, and contributes to the high accuracy of our predictive model.

In conclusion, this study represents a pivotal advancement toward leveraging a multifaceted approach encompassing both genetic and non-genetic determinants, including lifestyle choices, socioeconomic variables, and medication usage, to stratify individuals based on their likelihood to have skin cancer. While previous efforts have focused on utilizing genetics, such as polygenic risk scores, to categorize individuals according to disease susceptibility, our findings demonstrate the potential of integrating genetic insights with lifestyle and socioeconomic factors within expansive datasets comprising hundreds of thousands of individuals. Further enhancements may be achieved by integrating the approach outlined in this study with measurements of pertinent metabolites, such as plasma vitamin D levels (LOINC code 1989-3, assessed in 63,180 AoU participants), and polygenic scores for skin-relevant traits, such as skin pigmentation, which are not normally measured. Through this integrative approach, we are able to discern between individuals at elevated and reduced risk of having skin cancer, thereby offering valuable insights for early detection and personalized healthcare interventions. Ultimately, the approach proposed in this study may be applied to other diseases and conditions and, by integrating genetics, SDOH and lifestyle with other features present in the AoU database, such as lab measurements, procedures, drug prescriptions and wearable data, it may will possible to develop accurate models for multiple diseases.

## Methods
This study was conducted using the All Of Us (AoU) data freeze 7 at February 15, 2023. It contains the medical records and genetic information for 401,430 participants, including 154,123 males and 247,307 females.

### Data extraction and processing
The research presented here complies with the AoU ethical regulations.

Analyses were performed with R 4.4.0 on the AoU Researcher Workbench. We obtained each participant's general information, including person ID, self-reported gender, self-reported race and ethnicity, date of birth and death from the *person* and *death* tables in the AoU OMOP database.

### Cancer history.
Malignant tumor information was obtained by extracting all ICD9 tumor codes (140-209) and ICD10 tumor codes (C00-C80) from the *condition_occurrence* table. ICD9 codes 172 and

173, as well as ICD10 C43-C44 were used for "skin cancer". "Cancer history" was defined as being diagnosed with any non-skin cancer at a date earlier than skin cancer diagnosis.

### SDOH information.
SDOH variables were obtained from the "The Basics" survey in the *ds_survey* table. The variables were organized as follows: 1) annual income was divided into nine levels: <$10k; $10-25k; $25-35k; $35-50k; $50-75k; $75-100k; $100-150k; $150-200k; and > $200k; 2) insurance had yes/no values; 3) education (question: "What is the highest grade or year of school you completed?") had eight levels: never attended; one through four; five through eight; nine through eleven; twelve or GED; college one to three; college graduate; advanced degree; and 4) living status (question: "In the past 6 months, have you been worried or concerned about NOT having a place to live?")

### Lifestyle information.
Lifestyle, including alcohol consumption, and daily and yearly smoking were obtained from "Lifestyle" survey in the *ds_survey* table.

### UV light exposure.
The ZIP code associated with each person (query on the *observation* table with "where" clause *observation_source_value* = 'StreetAddress_PIIZIP'). Using the zipcodeR R package[45], we converted ZIP codes to latitude and longitude coordinates, which we used as a proxy for UV light exposure.

### Predicted ancestry.
Predicted ancestries were used as covariates for the prediction analyses. The AoU 401,430 participants were divided as follows: 218,799 s as European descent (54.5%), followed by African (75,654, 18.8%), Hispanic (72,977, 18.2%), Asian (13,633, 3.4%), Admixed (multiple ethnicities reported, 6608, 1.65%), Middle Eastern (2327, 0.60%) and Pacific Islanders (389, 0.01%). Each ancestry was coded as a binary variable.

### Genetic ancestry.
Coordinates of 16 genotype PCs, as well as predicted genetic ancestry proportions and genetic ancestry were obtained from the *srWGS* genetic predicted ancestry information table. The population definitions provided by AoU are based on the six superpopulations in the 1000 Genomes Project, with the addition of "Middle Eastern": European (EUR), African (AFR), American Admixed/Latino (AMR), East Asian (EAS), South Asian (SAS), and Middle Eastern (MID). All individuals that were not mapped to any of these six populations were labeled as "Other" (OTH).

### PDE5a inhibitor drug information.
Erectile dysfunction drug use was obtained from the *drug_exposure* table by querying all concepts that included the terms "Viagra", "Cialis", "Stendra", "sildenafil", "avanafil" or "tadalafil".

Part of the survey data was converted to binary: health insurance status; and living situation ("Living Situation: Stable House Concern"). The remaining variables obtained from surveys were converted to numeric: annual income; educational attainment; alcohol and smoking status.

Overall, 179,094 participants had complete data and were used for predictions.

### Statistical analyses
Packages bigrquery 1.5.1, tidyverse 2.0.0, plyr 1.8.9, dplyr 1.1.4, dbplyr 2.5.0 and data.table 1.15.2 were used in R 4.4.0 to process the queries from the AoU database and process the data.

We used Fisher's exact test for testing binary variables (*fisher.test* function) for association with skin cancer and t-tests for numeric variables (*t.test* function). Logistic regression was performed using the *glm* function in R. Cox proportional hazard models were computed with the *coxph* function in the survival 3.6–4 package in R: models were computed without adding any additional covariate.

We used t tests to investigate the differences in genotype PC coordinates between skin cancer patients and healthy individuals (i.e. not diagnosed with skin cancer).

We built a LASSO regression model using the *cv.glmnet* function from the glmnet 4.1.8 package in R: we used the genetic ancestry predictions probabilities, as provided by AoU, sex, age, SDOH (education, annual income and insurance status), cancer history, PDE5a inhibitor prescription, latitude, longitude and ancestry as covariates.

All p-values were 2 tailed, with the significance level set at p < 0.05. Corrections for multiple testing were performed using Benjamini-Hochberg's method with the *p.adjust* function.

SHAP analyses were performed using shapviz 0.9.7 and SHAP-forxgboost 0.1.3 in R.

Supplemental tables were generated with openxlsx 4.2.5.2 in R.

### Predictions

We selected XGBoost to predict skin cancer occurrence because it provides multiple advantages compared with other machine learning methods:

1. It models non-linear relationships between features and outputs;
2. It is highly efficient, making it suitable for datasets like AoU, which include hundreds of thousands of samples and may include hundreds of features;
3. It includes built-in Lasso and Ridge regularizations, helping prevent overfitting;
4. Its automatic feature selection allows prioritizing variables that may be strongly correlated with each other;
5. It does not require imputation of missing variables.

XGBoost predictions were performed in R using the *caret* 6.0-94 and *xgboost* 1.7.7.1 packages. First, we extracted 183,339 participants that had complete data (no NA values for any of the covariates) and divided the dataset into 80% training and 20% test. We trained 45 models using combinations of the five cancer categories (melanoma, SCC, BCC, other skin cancers and any skin cancer), three population groups (individuals of European ancestries, non-European ancestries and all participants) and sexes (males, females and both sexes). For each model, we proceeded to optimize XGBoost hyperparameters on the training set by controlling the *tuneGrid* parameter in the caret function *train* and performing a five-fold cross-validation at each step (function *trainControl* in caret). We optimized: maximum number of trees (50 to 200 in increments of 50); maximum tree depth (2, 4 or 6); and learning rate (0.1, 0.3 and 0.5). For each model, we selected the parameters that provided the best predictions (*bestTune* parameter) and evaluated their performance using each model's accuracy as summary metric (*metric = "Accuracy"* in the *caret::train* function). We obtained predictions on the training set for each model and used the *coords* function from the pROC package to determine the optimal threshold that maximizes both sensitivity and specificity. Logistic regression predictions were performed using the same training set, five-fold cross-validation and variables described for the XGBoost predictions with the function *glm* with parameter *family = binomial(link = "logit")*. Figures 5A, S4, S5, Table S5 and Table S6 show the F1 statistics and accuracy metrics calculated on the cross-validated training set.

To examine how variables contribute to prediction accuracy when training the XGBoost multiethnic model (any skin cancer, all participants and both sexes), we used the same process described above and datasets to train eight modifications of the multiethnic model by removing one set of variables (sex, age, cancer history, SDOH, lifestyle, genotype PCs, geo=longitude and latitude, PDE5a inhibitors) at a time (Figures S5, S6). We also calculated the gain and frequency of each variable in all the XGBoost trees using the *xgb.importance* function from the *xgboost* package.

Using the test set of participants who had complete information for all variables we calculated sensitivity, specificity, F1 statistic and NPV for all models (Fig. 5B, Table 1). We performed skin cancer prediction also on the 55,459 AoU participants who have genetic data but do not have SDOH information (Fig. 5C). We also examined the impact of missing variables in the "missing feature" individuals on the impact of the XGBoost multiethnic model when one variable (sex, age, cancer history, SDOH, lifestyle, genotype PCs, geo=longitude and latitude, PDE5a inhibitors) at a time was removed (Fig. 5D). In these cases, the XGBoost model handles missing information natively, therefore we did not perform any imputation.

### Reporting summary

Further information on research design is available in the Nature Portfolio Reporting Summary linked to this article.

## Data availability

All analyses were performed on the All of Us Researcher Workbench (https://www.researchallofus.org/data-tools/workbench/) in the "Impact of global and local ancestries on genome-wide association V7 studies" workspace (Controlled Tier data). As per AoU policies, we cannot share participant-level information or data on sets that include <20 participants. All other data is available as Supplemental Tables.

## Code availability

Code is available as Jupyter notebooks on Figshare: https://figshare.com/s/6c041849df0ef57eadad. The repository is licensed under the MIT License, an open-source license approved by the Open Source Initiative.

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

## Acknowledgements

This work was supported by the American Cancer Society IRG Grant IRG-19-230-48-IRG, UC San Diego Moores Cancer Center, Specialized Cancer Center Support Grant NIH/NCI P30CA023100, NIH/NHGRI grant RM1HG011558. W.G.G.R. was supported by Alfred P. Sloan Foundation's Minority Ph.D. (MPHD) Program [G-2022-10127] and by NIH grants T15LM011271. We gratefully acknowledge the All of Us participants, without whom this research would not have been possible. We also thank the National Institutes of Health's All of Us Research Program for providing access to the genotype and phenotype data used in this study. The authors thank John Charles Frazer for valuable discussions on including PDE5a inhibitors in the study design.

## Author contributions

K.A.F., M.D., M.G. and R.A.G. oversaw the study. M.D. and W.G.G.R. performed analysis. M.D. and K.A.F. conceived the study and wrote the manuscript.

## Competing interests

The authors declare no competing interests.
