## [Peer Review File · Nature Communications]

A highly accurate risk factor-based XGBoost multiethnic model for identifying patients with skin cancer

Corresponding Author: Dr Matteo D'Antonio

Editorial Note: Figure on page 10 in this Peer Review File has been amended to remove third-party material where no permission to publish could be obtained.

Version 0:

Reviewer comments:

Reviewer #1

(Remarks to the Author)

In this manuscript, D'Antonio and collaborators have built an XGBoost multiethnic machine learning model to identify associations between distinct factors (i.e. genetic, social determinants of health and lifestyle) and skin cancer risk. They claim they have built a model that can accurately predict skin cancer risk from a combination of these factors across distinct ancestries.

Overall, though interesting work, it was hard for this reviewer to understand exactly what the model is predicting. For example, it is surprising that age was not associated with skin cancer risk (line 181), as it is known that age associates with most types of cancer. However, later on (line 246) it is stated that age is the most important predictor of skin cancer on the XGBoost model. What is the reason for these conflicting results? The classification into ancestries also seems problematic (more details below). I believe it would greatly add to the clarity of this work if the findings from the model are spelled out in the manuscript, e.g. - cancer risk is associated with higher European ancestry, increasing age, higher SES, etc. This way, readers would be able to identify the combination of factors and association direction of the list of relevant covariables.

Major points

Line 69 - "Africans and Hispanics tend to be diagnosed with skin cancer at a significantly earlier age than Europeans" -- why would this be? This is a surprising finding which goes against what we currently know (in terms of the correlation between socioeconomic factors and ancestry (e.g., white people generally having better access to healthcare etc) and skin cancer risk. How was this analysis done? Can we see a graph of the ages at diagnosis of skin cancer for people of European descent vs all others? E.g. Does PC1 associate with the socioeconomic factors evaluated here too?

Line 89 - "Hispanic" is not an ancestry, it is an ethnicity. Hispanic people can be of any ancestry (e.g., see <https://doi.org/10.1371/journal.pgen.1002640> for a PCA with Hispanic people covering the span of ancestries). Please clarify what you mean with "Hispanic ancestry", as it usually would be contained within the Admixed group (but, again, some Hispanic people can have overly European or Native American ancestry). It is also surprising that Native American ancestry is not included (which, in itself is complicated as Native American ancestry includes many different ancestries).

How were people classified into ancestries - just chose the major ancestry for each person? If so, how was the Admixed group defined?

Line 169 - Was this analysis done on self-reported ancestries? I believe this would be better done with ancestries inferred from the PCA? How do the results change if so?

Given that age was the single most important predictor in the multiethnic, all included model, how does a model that only considers age fare?

Minor points

Introduction

Line 49 - I would add that these tumours are easily confused with other lesions such as diabetic ulcers, warts or fungal infections.

Line 59 - "The inhibitors downregulate PDE5A, which is encoded by PDE5A, a gene that degrades cyclic guanosine monophosphate..". I would remove "which is encoded by PDE5A", as the gene itself does not degrade cGMP but the protein does.

Line 66 - it would be good if the authors dedicated a few lines to explain what the XGBoost method is and its advantages.

Methods

Line 87 - Does self-reported race have a significant overlap with genotype PCA information?

Line 108 - "First, we extracted only participants that had a complete data (no NA values for any of the covariates)" - how many participants remained?

Line 183 - should this be Figure 2D?

Line 193 "Given the late detection rates of diagnosing skin cancer in non-European populations" - But earlier in the article it is stated that non-Europeans are diagnosed earlier. Where is this statement coming from?

Line 198 - How were the population groups defined, by self-report or by PCA?

Line 221 - "This analysis shows that the XGBoost model trained using any skin cancer, all participants, and skin cancer risk variables (sex, age, cancer history, SDOH, lifestyle and genetic background) predicts whether an individual has any type of skin cancer with high PPV and sensitivity for both Europeans and non-Europeans." - But the PPV is still substantially lower for non-Europeans than Europeans. Would the worse performance of the models trained on non-Europeans only be due to the number of participants available for the training? Or what could be the reason that a model trained on non-Europeans performs worse on non-Europeans than a model trained in Europeans?

Discussion

Line 288 - "These observations suggest that the genetic factors underlying skin cancer risk in Africans and Asians maybe different than the genetic factors associated with skin cancer in Europeans." This may be because, at least for melanoma, most of the skin cancer risk will be associated to UV in Europeans and non-UV factors in other populations. E.g., acral melanoma, not related to UV, is the most common type of melanoma in people from several ancestries different than European. Whether these other types of melanoma are related to genetics or are related to other factors remains to be determined.

Line 293 - "Wealthier individuals often exhibit a lower incidence of late-stage melanoma, attributed to heightened health awareness, and lifestyle,..". - but this is usually correlated with ancestry. Wealth is correlated with higher European ancestry in many countries. It would be good to see whether this is the case too in the All of Us database.

Figures

Figure 1A- Please clarify the legend as it is unclear at the moment. I suggest something like "Proportions of AoU participants of specific ancestries diagnosed....".

Figure 1G-H- I think it would be easier to interpret this figure if the association of European ancestry with PC1 was also shown.

Figure 2A-C. Please consider writing legends at the top of each graph (e.g., all participants, Europeans, non-Europeans) so it is more easily interpretable.

Figure 2D-F This legend seems to be wrong as 2D and 2E both say "Europeans". I assume D is "all participants"?

(Remarks on code availability)

I saw that the code is accessible and in a format that is easily run, however, I did not try to run it myself.

Reviewer #2

(Remarks to the Author)

D' Antonio et al, presented an interesting study in which they analyzed the impact of genetic and non-genetic factors on skin cancer risk. They used the All of Us dataset that includes information for more than 400,000 individuals in the USA with a prevalence of about 3.5% of skin cancer. They developed a multiethnic XGBoost model and compared its results with the ones obtained with a logistic regression model. Besides the generated model, the importance of the included variables was assessed to understand the most relevant factors that affect skin cancer risk. This analysis allowed an interpretation of the model thus contributing to a better understanding of diseases. Interestingly the study improved the prediction for all demographic groups. I find including genetic and environmental factors and using improved machine-learning models relevant for disease risk prediction in diverse groups. I have the following concerns:

- In Table S7 the referred value of 0.692 related to age is not shown, instead the age of diagnosis is listed (review line 250 from the manuscript). This is strange since one of the major findings of the study is the impact of age. In the code age_x seems to be defined as a vector, please clarify this.
- It is atypical to use 3 sets for validation since cross-validation is conducted, besides the other 2 sets (complete data and missing values). Commonly, the evaluation dataset is used to report the method's performance.
- The threshold to conduct the binary classification with optimal results should be mentioned.
- Other metrics such as the F1 score describe results better on an imbalanced dataset.
- Also, it is necessary to mention which summary metric is used to select the optimal model.
- I suggest analyzing the misclassified individuals to observe whether they share a common characteristic that leads to errors in the model. For example, I would suggest identifying them in the PCs.
- Authors should consider adding versions of the packages they use.
- The colors used in Figure S1 (aquamarine and cyan) are very similar and hard to distinguish.
- The caption from Figure 2D is incorrect. Check line 186 from the manuscript.
- In Figure 2, not all the dots that indicate the median and the significance are visible. However, the significance of PC3 is referenced in the text, line 186.

(Remarks on code availability)

Reviewer #3

(Remarks to the Author)

I co-reviewed this manuscript with one of the reviewers who provided the listed reports as part of the Nature Communications initiative to facilitate training in peer review and appropriate recognition for co-reviewers

(Remarks on code availability)

Reviewer #4

(Remarks to the Author)

The study developed multiethnic XGBoost models that integrate genetic and non-genetic factors to predict skin cancer risk with good accuracy. The research involved creating a large number of logistic regression and XGBoost models using various combinations of skin cancer categories, population groups, and sexes, providing a comprehensive view of aggregated and disaggregated machine learning experiments.

However, the manuscript does not include a comparison with other machine learning models previously published for skin cancer prediction. For a more rigorous evaluation, it would be beneficial to include a comparative analysis with other models from the literature to demonstrate the relative performance of the XGBoost model.

This study considered the demographic structure of the dataset, which is essential for machine learning fairness, and identified a significant performance gap between the EUR and non-EUR models (PPV = 0.904 vs PPV = 0.629). However, no mitigation measures were proposed to reduce this disparity.

Given that ~3.5% of the cohort has been diagnosed with skin cancer, the class imbalance is significant. It is unclear how this issue was handled or if it was addressed at all. The study used sensitivity, specificity, PPV, and NPV as performance metrics. These metrics depend on the classification threshold and are sensitive to class imbalance. To improve model evaluation, it would be beneficial to use the Area Under the Precision-Recall Curve (AUPR), which handles the trade-off between precision (PPV) and recall (sensitivity) across different threshold settings. The Precision-Recall curve is more informative in cases of class imbalance.

It seems that the XGBoost model is overly dependent on one risk factor -- age. Removing age from the input features resulted in a large decrease in PPV (from 0.904 to 0.097). The big drop in PPV suggests that the model heavily relies on age for making accurate predictions. Without this variable, the model's ability to correctly identify true positives significantly diminishes.

(Remarks on code availability)

Reviewer #5

(Remarks to the Author)

Summary:

The paper uses NIH All of Us (AoU) dataset to investigate the impact of genetics and various lifestyle factors on skin cancer risks for different races. The main findings include:

- Africans and Hispanics are diagnosed with skin cancer at an earlier age than Europeans;
- For self-reported African and Asian individuals, people with or without cancer have similar genetic backgrounds, likely

indicating existing genetic knowledge (extracted from European white) does not apply to them. The work uses XGBoost (an advanced decision tree model) and logistic regression models.

Strengths:

- The genetic aspect of the multiethnic analysis is new and interesting. Many existing machine learning publications on skin cancer focus on analyzing images. This work is different, as it examines genetic, demographic, sun exposure, related erectile dysfunction drugs, and lifestyle factors -- providing a refreshingly new perspective.
- The paper showcases how the NIH All of Us (AoU) dataset can be used for multi-ethnic analysis. The workflow described here could set a good example for other AOU studies.
- For mixed raced participants, the paper uses genetic information to identify the specific races, as opposed to relying on self-reported race and ethnicity information (which may not be accurate).

Weaknesses:

- The main weakness is that the paper seems to have two goals. One is the goal of understanding skin cancer risks of different demographic groups and the other is to show XGBoost is superior prediction model for all. While both goals are interesting, mixing them causes some confusion and distraction. It would be useful to clarify these two goals in the introduction and adjust the organization of the paper.
- Some of the claims regarding XGBoost's performance need to be adjusted.

Additional comments:

- Figure 1C about the age and cancer is difficult to read and needs to be explained more, as it is critical to the main finding of this paper, for example, how is X-axis calculated and how to interpret it.
- The feature ranking is usually easy to do for the decision tree type of machine learning models. So, it would be useful to add feature ranking analysis. In addition, the use of XGBoost (as opposed to other ML models) needs to be justified.
- The paper concludes XGBoost can predict skin cancer, e.g., "these results indicate that the general XGBoost model can predict skin cancer occurrence in all individuals". However, without comparison with other machine learning models or other baselines, this conclusion is not convincing. I suggest the authors rephrase these kinds of statements in the paper.
- The paper mentioned "Validation participants with missing data were predicted independently." This aspect needs to be elaborated, e.g., percentage of such participants with incomplete data, any data imputation methods used, and how it impacts the accuracy. (I understand Figure 3 has some analysis on missing data.)
- Can this workflow be generalized to analyze other diseases? Please share your insights.
- The paper is succinctly written and easy to follow.

(Remarks on code availability)

Reviewer #6

(Remarks to the Author)

I co-reviewed this manuscript with one of the reviewers who provided the listed reports as part of the Nature Communications initiative to facilitate training in peer review and appropriate recognition for co-reviewers.

(Remarks on code availability)

Version 1:

Reviewer comments:

Reviewer #1

(Remarks to the Author)

I am thankful to the authors for their thoughtful answers and consideration of the points I raised in the previous version of the manuscript. I have re-evaluated the manuscript in full and I believe it is much more understandable now, the added points and clarifications have made it easier to follow. I have no major concerns with the current version. I just have some minor comments that I think would further help make the manuscript clearer:

1. In the first paragraph of the introduction, perhaps clarify that the use of PDE5A inhibitors MAY be associated with skin cancer incidence? (There seems to be conflicting results in this issue). Similar in the sentence "Three erectile dysfunction drugs, sildenafil (Viagra), tadalafil (Cialis) and avanafil (Stendra), which inhibit PDE5a, have gained substantial attention due to their widespread use and association with skin cancer risk [9–12,14,21]." - perhaps just add "potential association with skin cancer".
2. Line 70 "In this study, to reduce current skin cancer outcome disparities" - Perhaps reword to "to investigate current skin cancer outcome disparities.."
3. I don't think Figure 1E shows that non-EUR Europeans tend to be diagnosed at an earlier age - could a simple comparison of median/IQR age of diagnosis among ethnicities be shown to make this point clearer?
4. Figures 3A-B. These are supposed to show that "OTH individuals who self-reported as European have a higher incidence and later age at diagnosis compared with other OTH individuals who self-reported as Admixed (multiple ancestries reported) or Hispanic", but the figures show the relationship of age at diagnosis and survival for the different categories. I would suggest making this simpler just to show the median and IQR of the age of diagnosis in different groups and list the incidences authors find. Similar comment with C-D.
5. Figure 1 legends- It would be clearer if in the legend, the full names of the populations were spelled out.
6. Please recheck Supplementary Table numbering. I think Table S1 appears to currently be Table S2, and perhaps this propagates.

(Remarks on code availability)

I did not attempt to run the code, but I downloaded it and read it and it appears well organised. Perhaps just a simple README would further clarify the order in which it has to be run and what inputs and outputs are expected in each notebook.

Reviewer #2

(Remarks to the Author)

Thank you for addressing many of my comments in this revised version of the manuscript.

Comment 1. The distinction between "age" and "age at diagnosis" remains unclear to me. The authors report a significant gain associated with the age at diagnosis, which seems intuitive given that controls would not have an age at diagnosis. However, I am unsure whether this refers to the age at participation for all individuals or specifically to the age at diagnosis for cases.

Comment 9. Figure 4 has improved in this version, but panel 4C is still cut off. Please ensure that the figure is fully visible and properly formatted in the final version.

Comment 2. I appreciate the authors' efforts to address my previous comment regarding dataset division. However, I remain concerned about the division into training and validation sets, especially since cross-validation is employed. In such cases, it is more appropriate to divide the dataset into training and testing sets, as the validation data is inherently part of the training process during cross-validation. That said, I understand this may be a matter of terminology, and I agree with the authors on having two divisions. Clarifying this point in the text would help avoid potential misunderstandings.

Another aspect I would like to point out is that the authors state that "cancer risk in admixed individuals is associated with the fraction of European ancestry in their genomes." While this is an interesting observation, it appears to be based primarily on PCA results. This is a strong claim, and I recommend that the authors support it further by incorporating admixture analysis or other ancestry inference methods to provide more robust evidence for this conclusion. Further, if making a claim about association with genetic ancestry, it is important to consider if the effect may be coming from other socioeconomic determinants of health that genetic ancestry is serving as a proxy for. It is important to model these if they are available (see for example Sohail et al Nature 2023), or at the very least discuss this as a potential source of the effect and important limitation.

Finally, I found a writing error in line 209.

(Remarks on code availability)

Reviewer #3

(Remarks to the Author)

(Remarks on code availability)

Reviewer #4

(Remarks to the Author)

I appreciate the authors' efforts to address the comments raised in the initial review. Below is an assessment of their revisions in response to each of the main points:

1. Comparison With Other Machine-Learning Models

The authors' explanation clarifies why a direct empirical comparison is impractical given the limitations of AoU data and the specific requirements of other models. They have also expanded the Introduction to acknowledge existing methods and their constraints. Overall, the discussion of existing tools is improved.

2. Addressing the Performance Gap Between EUR and non-EUR Groups

While the authors suggest that the multiethnic approach may improve performance for non-EUR groups, explicit fairness-focused measures remain underexplored. This is a limitation as the manuscript focuses on multiethnic models.

3. Class Imbalance and Performance Metrics (AUPR)

The authors stated, "We also include AUPR values in Table S5, Table S6, Table S7" but I did not find AUPR values in these tables or in the revised manuscript. While the inclusion of precision, recall, and F1 score improves model evaluation, these metrics only reflect performance at a single threshold. AUPR should be reported because it aggregates across all thresholds, offering a more robust, global evaluation of the model.

4. XGBoost Over-Reliance on Age

The XGBoost model is still overly dependent on one risk factor -- age. Removing age from the input features led to a substantial decline in F1 score (from 0.892 to 0.217). While the authors discussed the general reasons why XGBoost identifies age as a crucial predictor, they have not explicitly demonstrated how interactions between age and other factors influence predictions. Using SHAP interaction values could help elucidate these relationships. It is crucial to determine whether the model's strong reliance on age for its high performance is a valid outcome or an artifact of the modeling approach.

(Remarks on code availability)

Reviewer #5

(Remarks to the Author)

The revised paper and the itemized response file are both satisfactory. This work on admixed individuals addresses an important problem. The use of NIH All of Us datasets is relatively new. I recommend accepting the paper.

(Remarks on code availability)

Reviewer #6

(Remarks to the Author)

(Remarks on code availability)

Version 2:

Reviewer comments:

Reviewer #2

(Remarks to the Author)

Thank you for addressing many of my comments in the revised manuscript. I have two remaining suggestions to improve clarity and rigor.

Comment 1: Please report model performance on the test set in the text, not just cross-validation, please indicate which values correspond to the CV and the test. While CV is useful for model selection, final metrics should come from independent test data to properly assess generalizability and describe the model's performance.

Comment 2: I suggest minor revisions to clarify the discussion of genetic ancestry, ensuring terminology accurately reflects the continuous nature of human genetic variation and avoids potential misinterpretations. To improve precision, I recommend using terms like 'genetic similarity' rather than 'genomic composition' or 'fractions of ancestry.' For instance, but not exclusively:

Line 109: The phrasing "as expected" could be misinterpreted as continental ancestries being fixed biological boundaries. I

would avoid it and frame it differently.

Paragraphs 123-134: Please specify 'predicted continental ancestry' to emphasize that these are proxy labels based on similarity, not definitive or exhaustive categories.

(Remarks on code availability)

I did not run the code, but I did briefly check the notebooks. I think the documentation can be improved, and they should list all required libraries.

Reviewer #3

(Remarks to the Author)

(Remarks on code availability)

Reviewer #4

(Remarks to the Author)

The authors have effectively addressed the concerns regarding the model's strong reliance on age by demonstrating non-linear and conditional dependencies between age, genetics, and other variables through SHAP interaction analyses. I have no further comments.

(Remarks on code availability)

Reviewer #1 (Remarks to the Author):

Expert in skin cancer genetics and genomics, computational genomics, ancestry and ethnicity analysis

In this manuscript, D'Antonio and collaborators have built an XGBoost multiethnic machine learning model to identify associations between distinct factors (i.e. genetic, social determinants of health and lifestyle) and skin cancer risk. They claim they have built a model that can accurately predict skin cancer risk from a combination of these factors across distinct ancestries.

Overall, though interesting work, it was hard for this reviewer to understand exactly what the model is predicting. For example, it is surprising that age was not associated with skin cancer risk (line 181), as it is known that age associates with most types of cancer. However, later on (line 246) it is stated that age is the most important predictor of skin cancer on the XGBoost model. What is the reason for these conflicting results? The classification into ancestries also seems problematic (more details below). I believe it would greatly add to the clarity of this work if the findings from the model are spelled out in the manuscript, e.g. - cancer risk is associated with higher European ancestry, increasing age, higher SES, etc. This way, readers would be able to identify the combination of factors and association direction of the list of relevant covariables.

Thank you for the thoughtful comments. We have addressed each of these points and believe that it has greatly improved the manuscript.

We now spell out the findings of our manuscript in the Abstract and in summary statements at the end of each section in the Results.

Abstract “We show skin cancer risk in admixed individuals is associated with the fraction of European ancestry in their genomes, while for other non-Europeans genetic ancestry is not a major contributing factor. Age is not an independent risk factor, but its exclusion from the training set reduces XGBoost prediction accuracy, suggesting complex associations with other risk factors. XGBoost prediction accuracy is decreased by removal of genetic ancestry information but robust to missing SDOH and other risk factor information”.

We address the Reviewer’s other comments in detail in the Major points section below.

Major points

1) Line 69 - "Africans and Hispanics tend to be diagnosed with skin cancer at a significantly earlier age than Europeans" -- why would this be? This is a surprising finding which goes against what we currently know (in terms of the correlation between socioeconomic factors and ancestry (e.g., white people generally having better access to healthcare etc) and skin cancer risk. How was this analysis done? Can we see a graph of the ages at diagnosis of skin cancer for people of European descent vs all others? E.g. Does PC1 associate with the socioeconomic factors evaluated here too?

To characterize the differences in age at diagnosis between populations, we performed a survival analysis: we performed a Cox proportional hazard test (*coxph* function in the R *surv* package) to compare incidence and age at diagnosis across populations (Figure 1D-E). While it may be counterintuitive that non-EUR populations have an earlier age at diagnosis, several studies have shown this trend^{1,2} or no differences across populations³. In Figure 3A-D, we perform a deeper analysis on Admixed individuals (OTH category) and show that, even among these individuals, those that self-identify as European have a higher incidence and later age at diagnosis than Admixed individuals (OTH) that self-identify as other populations. Furthermore, we observe little or no difference between individuals of EUR genetic ancestry and OTH individuals who self-report as EUR (Figure 3C-D). To investigate the associations between socioeconomic status and skin cancer, we added Figure S2, where we show that wealthier European individuals are diagnosed more frequently and at a younger age than poorer European individuals.

We have substantially rewritten the Results section “Genetic and self-reported ancestry influences skin cancer risk” and added a new section “The proportion of EUR genomes influences skin cancer risk in OTH and AMR individuals” to describe these analyses.

Also see our response to Reviewer 5 comment 6.

2) Line 89 - "Hispanic" is not an ancestry, it is an ethnicity. Hispanic people can be of any ancestry (e.g., see <https://doi.org/10.1371/journal.pgen.1002640> for a PCA with Hispanic people covering the span of ancestries). Please clarify what you mean with "Hispanic ancestry", as it usually would be contained within the Admixed group (but, again, some Hispanic people can have overly European or Native American ancestry). It is also surprising that Native American ancestry is not included (which, in itself is complicated as Native American ancestry includes many different ancestries).

We agree that the definition of “Hispanic” was confusing in the original manuscript. In response to Reviewer 1, comment 3 we changed from using self-reported ancestry to genetic ancestry, and used the population definitions provided by AoU, which are based on the six superpopulations in the 1000 Genomes Project, with the addition of “Middle Eastern”: European (EUR), African (AFR), American Admixed/Latino (AMR), East Asian (EAS), South Asian (SAS), and Middle Eastern (MID). In the AoU, all individuals that did not map to any of the six populations were labeled as “Other” (OTH). We added two subsections “Self-reported ancestry” and “Genetic ancestry” to the Methods and have rewritten the Results section “Genetic and self-reported ancestry influences skin cancer risk” to address the reviewer’s comment.

We agree it is complicated to define Native American ancestry, and it is not reported in AoU in general (see below a snapshot of the self-reported ancestry and ethnicity from <https://www.researchallofus.org/data-tools/data-snapshots>): using the superpopulations definitions provided by the 1000 Genomes Project, there is no “Native American” superpopulation, but rather “Admixed American”, which is strongly associated with individuals who self-report as “Hispanic or Latino”.

[Figure Redacted]

3) How were people classified into ancestries - just chose the major ancestry for each person? If so, how was the Admixed group defined? Line 169 - Was this analysis done on self-reported ancestries? I believe this would be better done with ancestries inferred from the PCA? How do the results change if so?

We thank the reviewer for this insightful comment. In our original submission, we had annotated each individual's ancestry based on their self-reported race and ethnicity. In this revised submission, we changed to PCA-reported ancestry, as defined by All Of Us. We analyzed the correspondence between self-reported and PCA-based ancestry and ethnicity in Table S1 and in the Results section "Genetic and self-reported ancestry influences skin cancer risk". Based on the discrepancies between self-reported and genetic ancestries and knowing how skin cancer is strongly dependent on ancestry, we further investigated the ancestry of genetically Admixed individuals, and show that their ancestry composition is strongly associated with their self-reported ancestry, and how this influences skin cancer risk and age at diagnosis (Figure 2, Figure 3, Table S1, Table S2 and Table S3).

4) Given that age was the single most important predictor in the multiethnic, all included model, how does a model that only considers age fare?

Running an XGBoost model with only one covariate presents several limitations:

Overfitting Risk: XGBoost is a powerful algorithm that builds multiple decision trees in sequence, but with only one covariate, it has a limited number of ways to partition the data. This can lead to overfitting, where the model learns the noise rather than meaningful patterns.

Reduced Predictive Power: With a single covariate, the model's capacity to capture complex patterns and relationships in the data is limited. XGBoost typically performs best when it can use multiple covariates to identify interactions and nonlinear relationships, so a single covariate may limit its predictive performance.

Lack of Model Complexity Justification: XGBoost is generally most effective when there is a high-dimensional dataset with complex interactions to exploit. Using it on a dataset with only one feature could be overkill, as simpler models (e.g., linear

regression for regression tasks or logistic regression for binary classification) may perform equally well with less computational cost.

Potential for Inconsistent Results: XGBoost relies on parameters like learning rate, max depth, and regularization that work best with multiple covariates. With only one covariate, tuning these parameters may not yield meaningful improvements and could result in inconsistent or unreliable predictions.

In summary, for datasets with only one covariate, simpler models are often more appropriate, as they can provide similar performance with less complexity, lower risk of overfitting, and easier interpretability.

Above the reviewer also asks: “For example, it is surprising that age was not associated with skin cancer risk (line 181), as it is known that age associates with most types of cancer. However, later on (line 246) it is stated that age is the most important predictor of skin cancer on the XGBoost model. What is the reason for these conflicting results”

The conflicting results regarding age between XGBoost and logistic regression, can be explained by several factors:

Model Type and Complexity:

- XGBoost is a tree-based ensemble method that captures non-linear relationships between features and the target variable. It can automatically account for complex interactions between variables. This makes it more likely to prioritize variables like age that might have non-linear or interaction effects on the outcome. Age might be important in XGBoost if it interacts with other variables or has non-linear effects.
- Logistic Regression, on the other hand, is a linear model that assumes each predictor has a direct and additive effect on the log-odds of the outcome. If age does not have a linear relationship with the target, or its effect is moderated by other variables (i.e., interaction terms), logistic regression might not identify it as significant unless you explicitly include those non-linear terms or interactions.

Feature Importance Interpretation:

- In XGBoost, variable importance is measured by how often a feature is used to split the data and how much it improves the prediction accuracy. Thus, even subtle or complex relationships with the target can make a variable like age appear more important.
- In Logistic Regression, significance is determined by p-values, which rely on assumptions about the linear relationship between the predictor and the outcome. If the effect of age is small, linear regression might not flag it as significant, especially if other variables in the model explain more variation.

Multicollinearity:

- If age is correlated with other variables, logistic regression may assign the effect to those other correlated variables, reducing the apparent significance of age. In contrast, XGBoost can better handle multicollinearity due to its tree-based structure, which does not rely on independent variable assumptions.

Data Scaling and Feature Transformation:

- If age requires any transformation (e.g., quadratic or log transformation), it might not appear as significant.
- XGBoost, on the other hand, doesn't require feature scaling and can automatically handle non-linearity in the data.

In summary, the complexity and flexibility of XGBoost allow it to capture non-linear and interaction effects between age and other variables, which logistic regression might miss due to its linear nature and sensitivity to data scaling and multicollinearity.

We added the following explanation in the Discussion: “Furthermore, the discrepancy in the association with age between logistic regression and XGBoost can be explained by several factors, including: 1) model type and complexity: given that XGBoost is a tree-based ensemble method that captures non-linear relationships between features and the target variable, it accounts for complex interaction between variables, whereas logistic regression assumes that each predictor has a direct and additive effect on the log-odds of the outcome; 2) multicollinearity: if age is correlated with other variables, logistic regression may assign the effect to those other correlated variables, reducing the apparent significance of age; in contrast, XGBoost can better handle multicollinearity due to its tree-based structure, which does not rely on independent variable assumptions; and 3) data scaling and feature transformation: if age requires any transformation (e.g., quadratic or log transformation), it might not appear significant; XGBoost, on the other hand, doesn't require feature scaling and can automatically handle non-linearity in the data. The observation that missing age information while building the model results in the over-prediction of skin cancer in older individuals suggests that each additional risk factor contributes incrementally to prediction accuracy, providing information that is correlated with age. For example, there are strong associations between age and lifestyle factors, SDOH and using PDE5a inhibitors. Consequently, the absence of any single risk factor other than age does not significantly impair the model's predictive performance. This phenomenon sheds light on the fact that missing non-genetic data for validation participants does not substantially compromise prediction accuracy, as the model relies on a combination of statistically independent yet correlated risk factors to make accurate predictions. The fact that the XGBoost multiethnic model strongly depends on current age for accurate predictions limits its application for predicting future skin cancer risk, but makes it an ideal tool for clinicians to use to identify which patients should undergo whole-body skin cancer examines by trained dermatologists”. Please also see our response to Reviewer 4 comment 4.

Minor points

Introduction

5) Line 49 - I would add that these tumors are easily confused with other lesions such as diabetic ulcers, warts or fungal infections.

Thank you for the suggestion. We added the sentence “Furthermore, these tumors are easily confused with other lesions, including diabetic ulcers, warts or fungal infections, further complicating the diagnosis” in the Introduction.

6) Line 59 - "The inhibitors downregulate PDE5A, which is encoded by PDE5A, a gene that degrades cyclic guanosine monophosphate". I would remove "which is encoded by PDE5A", as the gene itself does not degrade cGMP but the protein does.

We updated the sentence to “Three erectile dysfunction drugs, sildenafil (Viagra), tadalafil (Cialis) and avanafil (Stendra), which inhibit PDE5a, have gained substantial attention due to their widespread use and association with skin cancer risk”.

7) Line 66 - it would be good if the authors dedicated a few lines to explain what the XGBoost method is and its advantages.

We agree that an explanation of the XGBoost model and its advantages is needed. We added to the Introduction: “eXtreme Gradient Boosting (XGBoost) is currently being used to predict COVID-19, stroke and other health conditions [28,29]. XGBoost was designed to improve both the performance and speed of machine learning models and offers multiple advantages compared to other machine learning methods, as it includes built-in Lasso and Ridge regularization, which prevent overfitting and improves model generalization, handles missing data and provides interpretable metrics such as feature importance scores allowing users to understand which features are most influential in the model’s predictions. Overall, XGBoost’s combination of speed, accuracy, flexibility, and robustness makes it a powerful choice to predict health outcomes”.

Methods

8) Line 87 - Does self-reported race have a significant overlap with genotype PCA information?

See response to Reviewer 1, Comment 3 above: we added **Table S2** to describe the correspondence between self-reported and genetic ancestry.

9) Line 108 - "First, we extracted only participants that had a complete data (no NA values for any of the covariates)" - how many participants remained?

We used 179,094 participants with complete data. This is the same set that we used for logistic regression in the Results section “Logistic regression models to predict the occurrence of skin cancer”. We changed the sentence to “Overall, 179,094 participants had complete data and were used for predictions”.

10) Line 183 - should this be Figure 4D?

We apologize for the mistake. We have updated the text and changed multiple figures. We have corrected this reference.

11) Line 193 "Given the late detection rates of diagnosing skin cancer in non-European populations" - But earlier in the article it is stated that non-Europeans are diagnosed earlier. Where is this statement coming from?

We apologize for the confusion: here we meant that skin cancer is usually detected at a more advanced and invasive stage in non-European populations. We updated the sentence to “Given that skin cancer is difficult to detect in non-European populations and hence detected at more advanced and invasive stages”.

12) Line 198 - How were the population groups defined, by self-report or by PCA?

In the current manuscript, population groups were defined by genetic ancestry, as provided by All Of Us, which used genotype PCA to annotate each individual with their most likely associated ancestry. To the models, we added self-reported ancestry as covariate. See response to Reviewer 1, comment 3.

13) Line 221 - "This analysis shows that the XGBoost model trained using any skin cancer, all participants, and skin cancer risk variables (sex, age, cancer history, SDOH, lifestyle and genetic background) predicts whether an individual has any type of skin cancer with high PPV and sensitivity for both Europeans and non-Europeans." - But the PPV is still substantially lower for non-Europeans than Europeans. Would the worse performance of the models trained on non-Europeans only be due to the number of participants available for the training? Or what could be the reason that a model trained on non-Europeans performs worse on non-Europeans than a model trained in Europeans?

We agree with the Reviewer's hypothesis: this difference is likely due to power, as a very small fraction of non-EUR individuals are diagnosed with skin cancer. When we combine all individuals, power is improved and prediction accuracy is increased also for non-EUR.

We describe this in the Introduction: "We trained XGBoost models only using non-European individuals and showed that they have low accuracy in predicting skin cancer patients (F1 mean statistic = 0.067) likely due to power issues"; and Results (section "XGBoost multiethnic model accurately predicts individuals with skin cancer"): "The poor performance of the non-EUR models was likely due to the fact that they were underpowered due to the relatively small number of non-EUR cancer patients".

Discussion

14) Line 288 - "These observations suggest that the genetic factors underlying skin cancer risk in Africans and Asians maybe different than the genetic factors associated with skin cancer in Europeans." This may be because, at least for melanoma, most of the skin cancer risk will be associated to UV in Europeans and non-UV factors in other populations. E.g., acral melanoma, not related to UV, is the most common type of melanoma in people from several ancestries different than European. Whether these other types of melanoma are related to genetics or are related to other factors remains to be determined.

We agree that the sentence pointed out by the Reviewer is confusing and does not take risk factors into account. We have rewritten the Discussion and removed this sentence.

15) Line 293 - "Wealthier individuals often exhibit a lower incidence of late-stage melanoma, attributed to heightened health awareness, and lifestyle,.."- but this is usually correlated with ancestry. Wealth is correlated with higher European ancestry in many countries. It would be good to see whether this is the case too in the All of Us database.

We agree that adding an analysis of wealth vs. skin cancer in European individuals would make the results stronger. Indeed, we found that wealthier European individuals are more likely to be diagnosed with skin cancer, are diagnosed earlier and, among European skin cancer patients, have increased survival rate. **Figure S2** describes the associations between annual income and skin cancer in the European population.

Figures

16) Figure 2A- Please clarify the legend as it is unclear at the moment. I suggest something like "Proportions of AoU participants of specific ancestries diagnosed....".

We updated the Figure legend (currently **Figure 1C**): “Proportions of AoU participants from indicated genetic ancestries diagnosed with each cancer category”.

17) Figure 2G-H- I think it would be easier to interpret this figure if the association of European ancestry with PC1 was also shown.

This is now **Figure 3F-G**. We have added the association of PC1 with EUR individuals as suggested. We added EUR also to **Figure S1** and **Table S3**.

18) Figure 4A-C. Please consider writing legends at the top of each graph (e.g., all participants, Europeans, non-Europeans) so it is more easily interpretable.

We added a legend at the top of each plot as suggested.

19) Figure 4D-F This legend seems to be wrong as 2D and 2E both say "Europeans". I assume D is "all participants"?

We apologize for the mistake and have corrected the legend.

Reviewer #1 (Remarks on code availability):

I saw that the code is accessible and in a format that is easily run, however, I did not try to run it myself.

Reviewer #2 (Remarks to the Author):

Expert in human genetics and genomics, ancestry and ethnicity analysis, computational and statistical genetics, and AI models

D' Antonio et al, presented an interesting study in which they analyzed the impact of genetic and non-genetic factors on skin cancer risk. They used the All of Us dataset that includes information for more than 400, 000 individuals in the USA with a prevalence of about 3.5% of skin cancer. They developed a multiethnic XGBoost model and compared its results with the ones obtained with a logistic regression model. Besides the generated model, the importance of the included variables was assessed to understand the most relevant factors that affect skin cancer risk. This analysis allowed an interpretation of the model thus contributing to a better understanding of diseases. Interestingly the study improved the prediction for all demographic groups. I find including genetic and environmental factors and using improved machine-learning models relevant for disease risk prediction in diverse groups. I have the following concerns:

1) In Table S6 the referred value of 0.692 related to age is not shown, instead the age of diagnosis is listed (review line 250 from the manuscript). This is strange since one of the major findings of the study is the impact of age. In the code age_x seems to be defined as a vector, please clarify this.

We thank the reviewer for pointing out this error. We have updated the tables and fixed this issue. In the code, “age_x” refers to a column that includes the age information.

2) It is atypical to use 3 sets for validation since cross-validation is conducted, besides the other 2 sets (complete data and missing values). Commonly, the evaluation dataset is used to report the method's performance.

We agree with the reviewer and have removed the test set. In the current manuscript, we use only two sets (training (80%), validation (20%) in addition to validation set that has missing values.

3) The threshold to conduct the binary classification with optimal results should be mentioned.

We have added the thresholds to the supplementary tables (Table S5, Table S6, Table S7).

4) Other metrics such as the F1 score describe results better on an imbalanced dataset.

We thank the reviewer for this important comment. We have substituted PPV with the F1 statistic in all figures and added precision, recall and F1 score in all supplemental tables.

5) Also, it is necessary to mention which summary metric is used to select the optimal model.

We added this information in the Methods section: “**For each model, we selected the parameters that provided the best predictions (*bestTune* parameter) and evaluated their performance using each model's accuracy as summary metric (*metric = “Accuracy” in the caret::train function*)**”.

6) I suggest analyzing the misclassified individuals to observe whether they share a common characteristic that leads to errors in the model. For example, I would suggest identifying them in the PCs.

We agree that understanding the associations between false positives and each feature would strengthen the results. Therefore, we ran a logistic regression with outcome being either true positives or false positives and used all the variables

from the XGBoost model as covariates. We found that, after FDR correction, only cancer history and age were significant, suggesting that these two variables influence the accuracy of the model.

We added **Table S9** and the following text to the Results section “Machine learning models to predict the occurrence of skin cancer”: “While the XGBoost multiethnic model had relatively few false negatives, there were hundreds of false positives. We explored whether false positive individuals exhibited distinct characteristics compared to those correctly classified as skin cancer patients. To do this, we conducted a logistic regression analysis in the EUR group, with the outcome categorized as either true positive (n = 1,644) or false positive (n = 353), using all variables from the XGBoost multiethnic model as covariates. The analysis revealed that most risk variables were not significantly different between the true positives and false positives; however, false positives were generally younger ($p = 3.7e-15$) and were less likely to have a history of cancer ($p = 2.1e-27$, **Table S9**). These findings suggest that younger individuals with many risk factors (excluding a history of cancer) tend to be incorrectly predicted to have skin cancer”.

7) Authors should consider adding versions of the packages they use.

We have added the package version in the Methods section.

8) The colors used in Figure S1 (aquamarine and cyan) are very similar and hard to distinguish.

We changed the color of Middle Eastern populations to dark purple.

9) The caption from Figure 4D is incorrect. Check line 186 from the manuscript.

We apologize for the mistake and have corrected the legend.

20) In Figure 4, not all the dots that indicate the median and the significance are visible. However, the significance of PC3 is referenced in the text, line 186.

We included the missing dots and added “Covariates with effect size >4 and < -4 were set to 4 and -4 respectively for visualization purposes” to the figure legend.

Reviewer #3 (Remarks to the Author):

Early Career Researcher co-reviewer

I co-reviewed this manuscript with one of the reviewers who provided the listed reports as part of the Nature Communications initiative to facilitate training in peer review and appropriate recognition for co-reviewers

Reviewer #4 (Remarks to the Author):

Expert in fair AI and deep learning for healthcare, and cancer genomics

The study developed multiethnic XGBoost models that integrate genetic and non-genetic factors to predict skin cancer risk with good accuracy. The research involved creating a large number of logistic regression and XGBoost models using various combinations of skin cancer categories, population groups, and sexes, providing a comprehensive view of aggregated and disaggregated machine learning experiments.

1) However, the manuscript does not include a comparison with other machine learning models previously published for skin cancer prediction. For a more rigorous evaluation, it would be beneficial to include a comparative analysis with other models from the literature to demonstrate the relative performance of the XGBoost model.

We thank the Reviewer for this comment and agree that we are missing a comparative analysis with previously published models. We have identified several relevant methods:

- Melanoma risk assessment tool (<https://mrisktool.cancer.gov/calculator.html>)⁴ predicts individual's risk based on: age; race/ethnicity; residency; gender; skin characteristics; history of sunburn; and physical exam, including mole and nevi presence and size, freckling and sun damage on the patient's back and shoulders.
- Disease risk score (DRS) and age-independent disease risk score (DRSA)⁵ uses 32 genetic and non-genetic risk factors to predict skin cancer lifetime risk using skin health and cancer-specific survey data (including questions about skin cancer family history, skin susceptibility and UV exposure) from 210,000 individuals.
- Naqvi et al.⁶ review more than 50 articles that use deep learning and AI with to detect skin cancer on imaging data.

While all these tools show good accuracy, they currently cannot be applied to the AoU dataset, for several reasons:

1. Imaging data is not available from AoU; or
2. They require skin-specific information (such as presence of nevi and moles, sun damage and freckles, and sun exposure), which are currently not available in AoU; or
3. In the case of the melanoma risk assessment tool, predictions are valid only in non-Hispanic White individuals.

AoU will in the future provide EHR notes, in addition to the information currently available, which may be useful to extract more specific skin information. Combining language models from EHR notes with the model described in this study will contribute to improving its accuracy.

In response to the reviewers comment, we have updated the Introduction: “Over the past 20 years, numerous prediction models for skin cancer have been developed, demonstrating the effectiveness of integrating various factors such as lifestyle choices (e.g., sun exposure), genetic predispositions (including family history and phenotypic traits like eye, hair, and skin color), and imaging techniques for identifying nevi, moles, freckles, and sun damage [25–27]. These models not only assist in identifying individuals at risk but, in the case of many imaging prediction tools, also facilitate the direct detection of skin cancer. However, despite their promise, these approaches necessitate substantial amounts of skin-specific data, particularly

concerning dermatological visits and imaging procedures. Consequently, they are not easily applicable to broader health databases, such as All of Us or the UK Biobank. Furthermore, they were trained and tested only on non-Hispanic White individuals [25–27]”.

2) This study considered the demographic structure of the dataset, which is essential for machine learning fairness, and identified a significant performance gap between the EUR and non-EUR models (PPV = 0.904 vs PPV = 0.629). However, no mitigation measures were proposed to reduce this disparity.

We agree that there are disparities in how the model works between EUR and non-EUR individuals, which are likely due to the differences in statistical power between these populations, as EUR are ~6.7 times more likely to develop skin cancer than other populations. However, our study shows that combining all individuals in a single model results in increased accuracy across populations.

We added to the Discussion: “We show that, by combining all individuals into a single model, prediction accuracy greatly improves for non-European individuals. However, prediction accuracy is impacted when genotype PCs are not included for individuals, which shows that genetic information plays an important role in the robust skin cancer predictions by the XGBoost multiethnic model”.

3) Given that ~3.5% of the cohort has been diagnosed with skin cancer, the class imbalance is significant. It is unclear how this issue was handled or if it was addressed at all. The study used sensitivity, specificity, PPV, and NPV as performance metrics. These metrics depend on the classification threshold and are sensitive to class imbalance. To improve model evaluation, it would be beneficial to use the Area Under the Precision-Recall Curve (AUPR), which handles the trade-off between precision (PPV) and recall (sensitivity) across different threshold settings. The Precision-Recall curve is more informative in cases of class imbalance.

To respond to this comment as well as comment 3 and 4 by Reviewer 2, we substituted PPV with F1 statistic in all figures (S, S3 and S4) and added precision, recall and F1 score in all supplemental tables. We also include AUPR values in Table S5, Table S6, Table S7.

4) It seems that the XGBoost model is overly dependent on one risk factor -- age. Removing age from the input features resulted in a large decrease in PPV (from 0.904 to 0.097). The big drop in PPV suggests that the model heavily relies on age for making accurate predictions. Without this variable, the model's ability to correctly identify true positives significantly diminishes.

We agree with the Reviewer: as with many cancer types, age is the major risk factor, as expected. Please see our response above to Reviewer 1, comment 4 where we discuss this in detail.

We discuss this more in depth in the Discussion: “Furthermore, while logistic regression analysis indicated that age was not independently associated with skin cancer occurrence, its exclusion from the training of our XGBoost multiethnic model led to a notable decline in prediction accuracy. This is in general expected in cancer studies, as age is a major risk factor for most cancers. The discrepancy in the association with age between logistic regression and XGBoost can be explained by several factors”.

Reviewer #5 (Remarks to the Author):

Expert in fair AI and deep learning for healthcare

Summary:

The paper uses NIH All of Us (AoU) dataset to investigate the impact of genetics and various lifestyle factors on skin cancer risks for different races. The main findings include:

- Africans and Hispanics are diagnosed with skin cancer at an earlier age than Europeans;*
- For self-reported African and Asian individuals, people with or without cancer have similar genetic backgrounds, likely indicating existing genetic knowledge (extracted from European white) does not apply to them.*

The work uses XGBoost (an advanced decision tree model) and logistic regression models.

Strengths:

1) The genetic aspect of the multiethnic analysis is new and interesting. Many existing machine learning publications on skin cancer focus on analyzing images. This work is different, as it examines genetic, demographic, sun exposure, related erectile dysfunction drugs, and lifestyle factors -- providing a refreshingly new perspective.

We thank the reviewer for this positive comment.

2) The paper showcases how the NIH All of Us (AoU) dataset can be used for multi-ethnic analysis. The workflow described here could set a good example for other AOU studies.

Thank you, we are currently working on other studies that exploit the multiethnic nature of AoU.

3) For mixed raced participants, the paper uses genetic information to identify the specific races, as opposed to relying on self-reported race and ethnicity information (which may not be accurate).

We agree that this is a point of strength for the manuscript, and added **Figure 2** to further investigate the associations between self-reported and genetic ancestry in relation to skin cancer risk. We also write in the Discussion: “**In this study, we adopted a global genotype PCA approach to incorporate genetic data into our skin cancer prediction model: by using this approach, there is no need to divide populations into distinct categorical variables, but we can analyze all participants, including Admixed individuals that frequently are excluded from genetic studies, in a single model**”.

Weaknesses:

4) The main weakness is that the paper seems to have two goals. One is the goal of understanding skin cancer risks of different demographic groups and the other is to show XGBoost is superior prediction model for all. While both goals are interesting, mixing them causes some confusion and distraction. It would be useful to clarify these two goals in the introduction and adjust the organization of the paper.

We updated the Abstract, Introduction, Results and Discussion sections to make these two goals clear.

For example in the Introduction we write: “In this study, to reduce current skin cancer outcome disparities because of diagnosis difficulty in non-Europeans we sought to develop an XGBoost model that can use risk factors to identify individuals with skin cancer regardless of their ancestry”.

5) Some of the claims regarding XGBoost's performance need to be adjusted.

We have rewritten most of the Abstract, Introduction and Discussion to tone down the claims about our XGBoost model.

Additional comments:

6) Figure 2C about the age and cancer is difficult to read and needs to be explained more, as it is critical to the main finding of this paper, for example, how is X-axis calculated and how to interpret it.

We agree that the original version of the figure was complicated to understand. Therefore, we have changed the figure to a survival analysis which provides a more standard visualization of the associations between age and cancer diagnosis in different populations.

See also comment 1 by Reviewer 1.

7) The feature ranking is usually easy to do for the decision tree type of machine learning models. So, it would be useful to add feature ranking analysis. In addition, the use of XGBoost (as opposed to other ML models) needs to be justified.

In response to the Reviewer’s comment, we have added the feature importance of the full model in **Table S8**. Age at diagnosis is the most important variable in the model, as expected since age is a major risk factor for skin cancer.

We describe our reasons for choosing XGBoost in the Methods section “Predictions”.

XGBoost provides multiple advantages compared with other machine learning methods:

1. It models non-linear relationships between features and outputs;
2. It is highly efficient, making it suitable for datasets like AoU, which include hundreds of thousands of samples and may include hundreds of features;
3. It includes built-in Lasso and Ridge regularizations, helping prevent overfitting;
4. Its automatic feature selection allows prioritizing variables that may be strongly correlated with each other;
5. It does not require imputation of missing variables.

8) The paper concludes XGBoost can predict skin cancer, e.g., “these results indicate that the general XGBoost model can predict skin cancer occurrence in all individuals”. However, without comparison with other machine learning models or other baselines, this conclusion is not convincing. I suggest the authors rephrase these kinds of statements in the paper.

We agree that this sentence is too strong and removed it from the manuscript.

9) The paper mentioned “Validation participants with missing data were predicted independently.” This aspect needs to be elaborated, e.g., percentage of such participants with incomplete data, any data imputation methods used, and how it impacts the accuracy. (I understand Figure 5 has some analysis on missing data.)

We agree that this statement is not clear. As XGBoost can treat missing values without the need for imputation, we performed predictions on the 55,459 participants who have genetic data but do not have SDOH information without imputing any of the missing variables.

We have changed the sentence to “We performed skin cancer prediction also on the 55,459 AoU participants who have genetic data but do not have SDOH information (Figure 5C)”.

10) Can this workflow be generalized to analyze other diseases? Please share your insights.

This is our intention in the long-term. We have added to the end of the Discussion: “Ultimately, the approach proposed in this study may be applied to other diseases and conditions and, by integrating genetics, SDOH and lifestyle with other features present in the AoU database, such as lab measurements, procedures, drug prescriptions and wearable data, it may will possible to develop accurate models for multiple diseases”.

11) The paper is succinctly written and easy to follow.

Thank you.

Reviewer #6 (Remarks to the Author):

Early Career Researcher co-reviewer

I co-reviewed this manuscript with one of the reviewers who provided the listed reports as part of the Nature Communications initiative to facilitate training in peer review and appropriate recognition for co-reviewers.

References

1. Cormier, J. N. *et al.* Ethnic Differences Among Patients With Cutaneous Melanoma. *Archives of Internal Medicine* **166**, 1907–1914 (2006).
2. Yuan, T.-A. *et al.* Race-, Age-, and Anatomic Site-Specific Gender Differences in Cutaneous Melanoma Suggest Differential Mechanisms of Early- and Late-Onset Melanoma. *International Journal of Environmental Research and Public Health* **16**, (2019).
3. Brady, J., Kashlan, R., Ruterbusch, J., Farshchian, M. & Moossavi, M. Racial Disparities in Patients with Melanoma: A Multivariate Survival Analysis. *Clinical, Cosmetic and Investigational Dermatology* **14**, 547 (2021).
4. Fears, T. R. *et al.* Identifying Individuals at High Risk of Melanoma: A Practical Predictor of Absolute Risk. *JCO* **24**, 3590–3596 (2006).
5. Fontanillas, P. *et al.* Disease risk scores for skin cancers. *Nat Commun* **12**, 160 (2021).
6. Naqvi, M., Gilani, S. Q., Syed, T., Marques, O. & Kim, H.-C. Skin Cancer Detection Using Deep Learning—A Review. *Diagnostics* **13**, 1911 (2023).

REVIEWER COMMENTS

Reviewer #1 (Remarks to the Author):

I am thankful to the authors for their thoughtful answers and consideration of the points I raised in the previous version of the manuscript. I have re-evaluated the manuscript in full and I believe it is much more understandable now, the added points and clarifications have made it easier to follow. I have no major concerns with the current version. I just have some minor comments that I think would further help make the manuscript clearer:

1. In the first paragraph of the introduction, perhaps clarify that the use of PDE5A inhibitors MAY be associated with skin cancer incidence? (There seems to be conflicting results in this issue). Similar in the sentence "Three erectile dysfunction drugs, sildenafil (Viagra), tadalafil (Cialis) and avanafil (Stendra), which inhibit PDE5a, have gained substantial attention due to their widespread use and association with skin cancer risk [9–12,14,21]." - perhaps just add "potential association with skin cancer".

We have updated the text as suggested.

2. Line 70 "In this study, to reduce current skin cancer outcome disparities" - Perhaps reword to "to investigate current skin cancer outcome disparities.."

We have updated the text as suggested.

3. I don't think Figure 1E shows that non-EUR Europeans tend to be diagnosed at an earlier age - could a simple comparison of median/IQR age of diagnosis among ethnicities be shown to make this point clearer?

We agree that survival analyses are not always straightforward to interpret. We have added **Figure S1** showing the boxplots of age at diagnosis for the different populations.

4. Figures 3A-B. These are supposed to show that "OTH individuals who self-reported as European have a higher incidence and later age at diagnosis compared with other OTH individuals who self-reported as Admixed (multiple ancestries reported) or Hispanic", but the figures show the relationship of age at diagnosis and survival for the different categories. I would suggest making this simpler just to show the median and IQR of the age of diagnosis in different groups and list the incidences authors find. Similar comment with C-D.

To add clarity to this analysis, we added **Figure S2** where we show the different skin cancer incidence and age at diagnosis in OTH individuals (A: barplot showing skin cancer incidence; and B: boxplot showing the distribution of ages at diagnosis) and in self-reported EUR individuals (C: barplot showing skin cancer incidence; and D: boxplot showing the distribution of ages at diagnosis)

5. Figure 1 legends- It would be clearer if in the legend, the full names of the populations were spelled out.

We updated the figure legend as suggested.

6. Please recheck Supplementary Table numbering. I think Table S1 appears to currently be Table S2, and perhaps this propagates.

We apologize for the mistake. We have updated the tables.

Reviewer #1 (Remarks on code availability):

I did not attempt to run the code, but I downloaded it and read it and it appears well organised. Perhaps just a simple README would further clarify the order in which it has to be run and what inputs and outputs are expected in each notebook.

To follow the Reviewer's advice, we added a README file in Figshare to help understand the different notebooks.

Reviewer #2 (Remarks to the Author):

Thank you for addressing many of my comments in this revised version of the manuscript.

Comment 1. The distinction between "age" and "age at diagnosis" remains unclear to me. The authors report a significant gain associated with the age at diagnosis, which seems intuitive given that controls would not have an age at diagnosis. However, I am unsure whether this refers to the age at participation for all individuals or specifically to the age at diagnosis for cases.

The Reviewer is correct. We agree it is unclear how we define age and age at diagnosis in the different analyses and that defining age in different ways between skin cancer patients and healthy individuals may cause issues in the predictions. To show that this does not affect the XGBoost model, we added to the Results section "Non-linear associations between age and other variables influence multiethnic predictions": "We also tested whether this strong reliance of the model on age is due to the fact that this variable is defined in different ways for skin cancer patients (age at diagnosis) and other individuals (current age). We built a model using current age for all individuals and obtained very high F1 statistic (0.909, data not shown)."

We have also added to the legends for **Figure 1D, Figure 3A,C**: "X axis indicates age at diagnosis for skin cancer patients; age at last follow-up for all other individuals".

Comment 9. Figure 4 has improved in this version, but panel 4C is still cut off. Please ensure that the figure is fully visible and properly formatted in the final version.

We have fixed the Figure to have panel C fully visible.

Comment 2. I appreciate the authors' efforts to address my previous comment regarding dataset division. However, I remain concerned about the division into training and validation sets, especially since cross-validation is employed. In such cases, it is more appropriate to divide the dataset into training and testing sets, as the validation data is inherently part of the training process during cross-validation. That said, I understand this may be a matter of terminology, and I agree with the authors on having two divisions. Clarifying this point in the text would help avoid potential misunderstandings.

We agree that the distinction between training and validation is not clear. As per the Reviewer's suggestion, we have changed "validation" to "test" set.

Another aspect I would like to point out is that the authors state that "cancer risk in admixed individuals is associated with the fraction of European ancestry in their genomes." While this is an interesting observation, it appears to be based primarily on PCA results. This is a strong claim, and I recommend that the authors support it further by incorporating admixture analysis or other ancestry inference methods to provide more robust evidence for this conclusion. Further, if making a claim about association with genetic ancestry, it is important to consider if the effect may be coming from other socioeconomic determinants of health that genetic ancestry is serving as a proxy for. It is important to model these if they are available (see for example Sohail et al Nature 2023), or at the very least discuss this as a potential source of the effect and important limitation.

We agree that adding a direct analysis of the associations between the proportions of genetic EUR ancestry and skin cancer risk when taking other lifestyle and SDOH variable into account would strengthen our conclusions. We performed a LASSO regression analysis on all OTH individuals and show that genetic the EUR proportion variable is important to assess the associations with skin cancer risk, even when accounting for lifestyle and SDOH.

We have added **Figure 3H**, updated the Results section “The proportion of EUR genomes influences skin cancer risk in OTH and AMR individuals”: “The variables with the strongest influence on skin cancer risk were cancer history and prescription of sildenafil and avanafil, followed by genetic EUR ancestry, self-reported Middle Eastern and self-reported European ancestry (**Figure 3H**). These results show that there is a strong linear association between the proportion of EUR ancestry and skin cancer in OTH individuals even when lifestyle and SDOH are taken into account”; and the Methods section “Statistical analyses”: “We built a LASSO regression model using the *cv.glmnet* function from the *glmnet* 4.1.8 package in R: we used the proportions of genetic ancestries, as provided by AoU, sex, age, SDOH (education, annual income and insurance status), cancer history, PDE5a inhibitor prescription, latitude, longitude and self-reported ancestry as covariates”. We have also added a reference to Sohail et al Nature 2023 in the Discussion: “Admixed individuals who self-report as Europeans have a higher age at diagnosis than individuals with European genetic ancestry, even when accounting for lifestyle and SDOH. Our results confirm the findings from previous studies [39], which demonstrated the importance of adjusting for environmental factors when investigating disease risk in diverse and admixed populations”.

Finally, I found a writing error in line 209.

Thank you, we fixed the mistake.

Reviewer #3 (Remarks to the Author):

Reviewer #4 (Remarks to the Author):

I appreciate the authors' efforts to address the comments raised in the initial review. Below is an assessment of their revisions in response to each of the main points:

1. Comparison With Other Machine-Learning Models

The authors' explanation clarifies why a direct empirical comparison is impractical given the limitations of AoU data and the specific requirements of other models. They have also expanded the Introduction to acknowledge existing methods and their constraints. Overall, the discussion of existing tools is improved.

Thank you.

2. Addressing the Performance Gap Between EUR and non-EUR Groups

While the authors suggest that the multiethnic approach may improve performance for non-EUR groups, explicit fairness-focused measures remain underexplored. This is a limitation as the manuscript focuses on multiethnic models.

We agree that further explanations are necessary. We added to the Discussion: “While overall our model shows improvements in predicting individuals of genetic non-EUR ancestry, further studies should be performed to further characterize other potential biases and ensure model fairness”.

3. Class Imbalance and Performance Metrics (AUPR)

The authors stated, “We also include AUPR values in Table S5, Table S6, Table S7” but I did not find AUPR values in these tables or in the revised manuscript. While the inclusion of precision, recall, and F1 score improves model evaluation, these metrics only reflect performance at a single threshold. AUPR should be reported because it aggregates across all thresholds, offering a more robust, global evaluation of the model.

We apologize for this. We have added ROC AUC and AUC PR to **Table S5**, **Table S6** and **Table S7**.

4. XGBoost Over-Reliance on Age

The XGBoost model is still overly dependent on one risk factor -- age. Removing age from the input features led to a substantial decline in F1 score (from 0.892 to 0.217). While the authors discussed the general reasons why XGBoost identifies age as a crucial predictor, they have not explicitly demonstrated how interactions between age and other factors influence predictions. Using SHAP interaction values could help elucidate these relationships. It is crucial to determine whether the model's strong reliance on age for its high performance is a valid outcome or an artifact of the modeling approach.

We agree that a deeper investigation of the effects of age on the XGBoost models is necessary. To answer to this comment, we have expanded the last part of the manuscript and added a deep analysis of the non-linear associations between age and the other variables. We have added a new Results section “Age and genotype PCs have strong non-linear associations with the prediction model” describing SHAP scores and SHAP interactions between variables, which show: 1) a non-linear association between age and skin cancer; and 2) different associations between other variables (in particular cancer history, genetic ancestry and annual income), age and skin cancer. This new analysis indicates non-additive relationships, where the joint influence of different features differs from the sum of their independent effects, highlighting conditional dependency

between variables. These results also confirm that age and genetics have strong non-linear effects on predictions, which make them important respectively for building the model and for accurate predictions.

We have added **Figure 6** and **Figure S7** showing the SHAP plots with feature importance and dependence of the XGBoost model on age. The plots show that age is the strongest predictor of skin cancer. They also show that there is a non-linear relationship between age and skin cancer occurrence, which explains the absence of an association between these two variables in the logistic regression model (**Figure 4D-F**).

Reviewer #5 (Remarks to the Author):

The revised paper and the itemized response file are both satisfactory. This work on admixed individuals addresses an important problem. The use of NIH All of Us datasets is relatively new. I recommend accepting the paper.

Thank you!

Reviewer #6 (Remarks to the Author):

Reviewer #2 (Remarks to the Author)

Thank you for addressing many of my comments in the revised manuscript. I have two remaining suggestions to improve clarity and rigor.

Comment 1: Please report model performance on the test set in the text, not just cross-validation, please indicate which values correspond to the CV and the test. While CV is useful for model selection, final metrics should come from independent test data to properly assess generalizability and describe the model's performance.

Thank you, we have added the model performance on both cross-validation and validation sets in the text (Results section “XGBoost multiethnic model accurately predicts individuals with skin cancer”).

Comment 2: I suggest minor revisions to clarify the discussion of genetic ancestry, ensuring terminology accurately reflects the continuous nature of human genetic variation and avoids potential misinterpretations. To improve precision, I recommend using terms like 'genetic similarity' rather than 'genomic composition' or 'fractions of ancestry.' For instance, but not exclusively:

Line 109: The phrasing “as expected” could be misinterpreted as continental ancestries being fixed biological boundaries. I would avoid it and frame it differently.

We have removed “as expected”.

Paragraphs 123-134: Please specify 'predicted continental ancestry' to emphasize that these are proxy labels based on similarity, not definitive or exhaustive categories.

We have changed “predicted genetic ancestry” to “predicted continental ancestry”.

(Remarks on code availability)

I did not run the code, but I did briefly check the notebooks. I think the documentation can be improve, and they should list all required libraries.

We have added all required libraries and a copy of `sessionInfo()` to the README.

Reviewer #3 (Remarks to the Author)

Thank you.

Reviewer #4 (Remarks to the Author)

The authors have effectively addressed the concerns regarding the model's strong reliance on age by demonstrating non-linear and conditional dependencies between age, genetics, and other variables through SHAP interaction analyses. I have no further comments.

Thank you.